



# Universal Differential Equations for glacier ice flow modelling

Jordi Bolibar[1,2,†], Facundo Sapienza[3,†], Fabien Maussion[4], Redouane Lguensat[5], Bert Wouters[1,2], and Fernando Pérez[3]

[1]Institute for Marine and Atmospheric research Utrecht, Utrecht University, Utrecht, Netherlands
[2]Faculty of Civil Engineering and Geosciences, Technische Universiteit Delft, Delft, Netherlands
[3]Department of Statistics, University of California, Berkeley, CA, USA
[4]Department of Atmospheric and Cryospheric Sciences, University of Innsbruck, Innsbruck, Austria
[5]Institut Pierre-Simon Laplace, IRD, Sorbonne Université, Paris, France
[†]These authors contributed equally to this work.

**Correspondence:** Jordi Bolibar (j.bolibar@uu.nl), Facundo Sapienza (fsapienza@berkeley.edu)

**Abstract.** Geoscientific models are facing increasing challenges to exploit growing datasets coming from remote sensing. Universal Differential Equations (UDEs), aided by differentiable programming, provide a new scientific modelling paradigm enabling both complex functional inversions to potentially discover new physical laws and data assimilation from heterogeneous and sparse observations. We demonstrate an application of UDEs as a proof of concept to learn the creep component
of ice flow, i.e. a nonlinear diffusivity differential equation, of a glacier evolution model. By combining a mechanistic model based on a 2D Shallow Ice Approximation Partial Differential Equation with an embedded neural network, i.e. a UDE, we can learn parts of an equation as nonlinear functions that then can be translated into mathematical expressions. We implemented this modelling framework as ODINN.jl, a package in the Julia programming language, providing high performance, source-to-source automatic differentiation (AD) and seamless integration with tools and global datasets from the Open Global Glacier
Model in Python. We demonstrate this concept for 17 different glaciers around the world, for which we successfully recover a prescribed artificial law describing ice creep variability by solving ~500,000 Ordinary Differential Equations in parallel. Furthermore, we investigate which are the best tools in the scientific machine learning ecosystem in Julia to differentiate and optimize large nonlinear diffusivity UDEs. This study represents a proof of concept for a new modelling framework aiming at discovering empirical laws for large-scale glacier processes, such as the variability of ice creep and basal sliding for ice flow,
and new hybrid surface mass balance models.

## 1 Introduction

In the past decade, remote sensing observations have sparked a revolution in scientific computing and modeling within Earth sciences, with a particular impact on the field of glaciology (Hugonnet et al., 2020; Millan et al., 2022). This revolution is assisted by modelling frameworks based on machine learning (Rasp et al., 2018; Jouvet et al., 2021), computational scientific
infrastructure (e.g. Jupyter and Pangeo (Thomas et al., 2016; Arendt et al., 2018)) and modern programming languages like Julia (Bezanson et al., 2017; Strauss et al., 2023). Machine learning methods have opened new avenues for extending traditional physical modeling approaches with rich and complex datasets, offering advances in both computational efficiency and

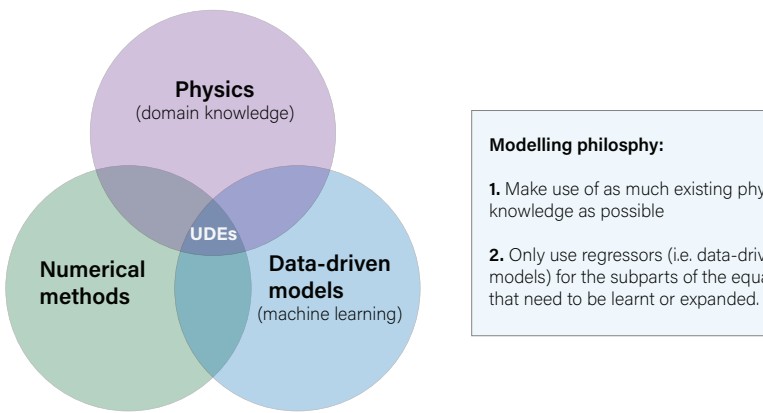

**Figure 1.** Basic representation of Universal Differential Equations (UDEs) and their associated modelling philosphy. UDEs sit at the intersection of physical domain knowledge, represented by differential equations, numerical methods used to solve the differential equations and data-driven models, often represented as machine learning.

predictive power. Nonetheless, the lack of interpretability of some of these methods, including artificial neural networks, has been a frequent subject of concern when modelling physical systems (Zdeborová, 2020). This "black box" effect is particularly

limiting in scientific modelling, where the discovery and interpretation of physical processes and mechanisms is crucial to both improving knowledge on physics and improving predictions. As a consequence, a new breed of machine learning models has appeared in the last years, attempting to add physical constraints and interpretability to learning algorithms (Raissi et al., 2017; Chen et al., 2019; Rackauckas et al., 2020).

Universal Differential Equations (UDEs, Rackauckas et al. (2020)), also known as Neural Differential Equations when

using neural networks (Chen et al., 2019; Lguensat et al., 2019; Kidger, 2022), combine the physical simulation of a differential equation using a numerical solver with machine learning (Figure 1). Optimization algorithms based on exact calculations of the gradient of the loss function require a fully differentiable framework, which has been a technical barrier for some time. Python libraries such as PyTorch (Paszke et al., 2019), Tensorflow (Abadi et al., 2016) or JAX (Bradbury et al., 2020) require rewriting the scientific model and the solver with the specific differentiable operations of each library, making it very costly to apply it

to existing models or making solutions very library-centered. Alternatively, the Julia programming language (Bezanson et al., 2017), designed specifically for modern scientific computing, has approached this problem in a different manner. Instead of using library-specific differentiable operators, it performs automatic differentiation (AD) directly on source code. This feature, together with a rich differential equations library (Rackauckas and Nie, 2017) provides a suitable scientific machine learning ecosystem to explore new ways to model and understand physical systems (Rackauckas et al., 2019).

In glaciology, models have not escaped these general trends. As for machine learning in general, classification methods have been more popular than regression methods (e.g. Baumhoer et al. (2019); Mohajerani et al. (2019)). Nonetheless, progress has been made with surrogate models for ice flow modelling (Riel et al., 2021; Jouvet et al., 2021), subglacial processes





(Brinkerhoff et al., 2020), glacier mass balance modelling (Bolibar et al., 2020a, b; Anilkumar et al., 2022; Guidicelli et al., 2023) or super-resolution applications to downscale glacier ice thickness (Leong and Horgan, 2020). In terms of modelling

glacier processes regionally or globally, it is still very challenging to move from small-scale detailed observations and physical processes to large-scale observations and parametrizations. When modelling glaciers globally, simple empirical models such as temperature-index models are used, due to their robustness to noise and the lack of observations needed to support more complex models. The same applies for ice flow dynamics, with flowline models based on the Shallow Ice Approximation (SIA, Hutter (1983)) being widely used, as it provides a good approximation, particularly with noisy and coarse-resolution

input data typical from large-scale models (Maussion et al., 2019; Zekollari et al., 2019). Moreover, it also helps to reduce the computational cost of simulations with respect to higher-order models. Therefore, there is a broad need for new methods enabling a robust calibration and discovery of more sophisticated, nonlinear interpretable parametrizations, in geosciences in general, but also for both glacier mass balance and ice flow dynamics. These include the need to transition towards non-constant melt and accumulation factors for temperature-index mass balance models (Bolibar et al., 2022), or the need to find a robust

relationship to calibrate ice creep and basal sliding for different regions, topographies and climates (Hock et al., 2023).

In terms of data assimilation and model parameter calibration, many different approaches to obtain differentiable glacier models have been developed (Morlighem et al., 2013; Goldberg and Heimbach, 2013; Brinkerhoff et al., 2016). These inverse modelling frameworks enable the minimization of a loss function by finding the optimal values of parameters via their gradients. Nonetheless, all efforts so far have been applied to the inversion of scalar parameters, i.e. parameters that are stationary

for a single inversion given a dataset. This means that the potential of learning the underlying physical processes is reduced to the current structure of the mechanistic model. To advance beyond scalar parameter inversions, more complex inversions are required, shifting towards functional inversions. Functional inversions enable the capture of relationships between a parameter of interest and other proxy variables, resulting in a function that can serve as a law or parametrization.

We present an application of Universal Differential Equations, i.e. a differential equation with an embedded function approx-

imator (e.g. a neural network). For the purpose of this study, this NN is used to infer a prescribed artificial law determining the ice creep coefficient in Glen's law (Cuffey and Paterson, 2010) based on a climate proxy. Instead of treating it as classical inverse problem, where a global parameter is optimized leading to a stationary parameter in space and time, neural networks learn a nonlinear function that captures the spatiotemporal variability of that parameter. This opens the door to a new way of learning parametrizations and empirical laws of physical processes from data. This case study is based on `ODINN.jl` v0.2 (Bolibar

and Sapienza, 2023), a new open-source Julia package, available on GitHub at https://github.com/ODINN-SciML/ODINN.jl. With this study, we attempt to share and discuss what are the main advances and difficulties in applying UDEs to more complex physical problems, we assess the current state of differentiable physics in Julia, and we suggest and project the next steps in order to scale this modelling framework to work with large scale remote sensing datasets.





## 2 Methods

In this section we introduce the Partial Differential Equation (PDE) describing the flow of ice through the SIA equation, and we present its equivalent Universal Differential Equation (UDE) with an embedded neural network.

### 2.1 Glacier ice flow modelling

We will consider the SIA equation to describe the temporal variation of the ice thickness (Cuffey and Paterson, 2010). Assuming a small depth-to-width ratio in ice thickness and that the driving stress caused by gravity is only balanced by the basal resistance,

the evolution of the ice thickness $H(x, y, t)$ can be described by

$$\frac{\partial H}{\partial t} = \dot{b} + \nabla \cdot \left( \left( C + \frac{2A}{n+2}H \right) (\rho g)^n H^{n+1} \|\nabla S\|^{n-1} \nabla S \right), \tag{1}$$

where $n$ and $A$ are the creep exponent and parameter in Glen's Law, respectively; $\dot{b}$ is the surface mass balance (SMB); $C$ is a basal sliding coefficient; and $\nabla S$ is the gradient of the glacier surface $S(x, y, t) = B(x, y) + H(x, y, t)$, with $B(x, y)$ the location of the bed, and $\|\nabla S\|$ denotes its Euclidean norm[1]. A convenient simplification of the SIA equation is to assume

$C = 0$, which implies a basal velocity equal to zero all along the bed. This is reasonable when large portions of the glacier bed experience minimal sliding. In that case, the SIA equation reduces to the diffusivity equation

$$\frac{\partial H}{\partial t} = \dot{b} + \nabla \left( D \nabla S \right), \tag{2}$$

with $D$ the diffusivity given by

$$D = \Gamma H^{n+2} \|\nabla S\|^{n-1}, \quad \Gamma = \frac{2A}{n+2}(\rho g)^n. \tag{3}$$

Except for a few engineered initial glacier conditions, no analytical solutions for the SIA equation exist and it is necessary to use numerical methods (Halfar, 1981).

    Importantly for our analysis, some of the coefficients that play a central role in the ice flow dynamics of a glacier (e.g. $A$, $n$ and $C$) are generally non constant and may vary both temporally and spatially. Although it is usually assumed that $n \approx 3$, this number can vary between 1.5 and 4.2 for different ice and stress configurations. Furthermore, the viscosity of the ice

and consequently the Glen parameter $A$ are affected by multiple factors, including ice temperature, pressure, water content, and ice fabric (Cuffey and Paterson, 2010). For example, ice is harder and therefore more resistant to deformation at lower temperatures. The parameters $A$, $n$ and $C$ are usually used as tuning parameters and may or may not vary between glaciers depending on the calibration strategy and data availability.

    An important propriety of the SIA equation is that the ice surface velocity $V$ can be directly derived from the ice thickness

$H$ by the equation

$$V = -\frac{2A}{n+1}(\rho g)^n H^{n+1} \|\nabla S\|^{n-1} \nabla S. \tag{4}$$

---

[1]Gradients here refer always to the spatial derivatives in $x$ and $y$.





Notice that the velocity field is a two dimensional vector that can be evaluated at any point in the glacier and points to the maximum direction of decrease in surface slope.

## 2.2 Universal Differential Equations

In the last years there has been an increasing interest in transitioning physical models to a data-driven domain, where unknowns in the laws governing the physical system of interest are identified via the use of machine learning algorithms. The philosophy behind Universal Differential Equations is to embed a rich family of parametric functions inside a differential equation, so the base structure of the differential equation is preserved but more flexibility is allowed in the model at the moment of fitting observed data. This family of functions, usually referred as the universal approximator because of their ability to approximate a large family of functions, includes among others, neural networks, polynomial expansions and splines. An example of this is a Universal Ordinary Differential Equation (Rackauckas et al., 2020)

$$\frac{du}{dt} = f(u, t, U_\theta(u, t)), \tag{5}$$

where $f$ is a known function that describes the dynamics of the system; and $U_\theta(u, t)$ represents the universal approximator, a function parametrized by a vector parameter $\theta$ that takes as an argument the solution $u$ and time $t$, as well as other parameters of interest to the physical system.

In this study, the function $f$ to be simulated is the SIA (equation (2)). Training such a UDE requires that we optimize with respect to the solutions of the SIA equation, which need to be solved using sophisticated numerical methods. The approach to fit the value of $\theta$ is to minimize the squared error between the observed ice surface velocity profile (described in section 3.1 together with all other datasets used) at some given time and the predicted surface velocities using the UDE, an approach known as trajectory matching (Ramsay and Hooker, 2017). For a single glacier, if we observed two different ice surface velocities $V_0$ and $V_1$ at times $t_0$ and $t_1$, respectively, then we want to find $\theta$ that minimizes the discrepancy between $V_1$ and $\text{SIASolver}(V_0, t_0, t_1, \theta)$, the numerical solution of the SIA equation embedded with the universal approximator $D_\theta$. When training with multiple glaciers, we are instead interested in minimizing the total level of agreement between observation and predictions,

$$\min_\theta \mathcal{L}(\theta) = \sum_k \omega_k \mathcal{L}_k(\theta), \qquad \mathcal{L}_k(\theta) = \|V_1^k - \text{SIASolver}(V_0^k, t_0, t_1, D_\theta)\|_F^2, \tag{6}$$

where $\|\cdot\|_F$ denotes the Frobenius norm, that is, the square root of the sum of the squares of all matrix entries; and each $k$ corresponds to a different glacier. The weights $\omega_k$ are included in order to balance the contribution of each glacier to the total loss function. For our experiments, we consider $\omega_k = 1/\|V_0^k\|_F$, which results in a re-scaling of the surface velocities. This scaling is similar to the methods suggested in Kim et al. (2021) to improve the training of stiff neural ODEs.

## 2.3 Functional inversion

We consider a simple synthetic example where we fix $n = 3$ - a common assumption in glaciology, $C = 0$, and model the dependency of Glen's creep parameter $A$ and the climate temperature normal $T$, i.e. the average long-term variability of air



temperature at the glacier surface. $T$ is computed using a 30-year rolling mean of the air temperature series, used to drive the changes in $A$ in the prescribed artificial law. Although simplistic and incomplete, this relationship allows us to present all of

our methodological steps in the process of identifying more general phenomenological laws for glacier dynamics. Any other proxies of interest could be used instead of $T$ for the design of the model. Instead of considering that the diffusivity $D_\theta$ is the output of a universal approximator, we are going to replace the creep parameter $A$ in Equation (3) with a neural network with input $T$:

$$D_\theta(T) = \frac{2\,A_\theta(T)}{n+2}(\rho g)^n\,H^{n+2}\|\nabla S\|^{n-1}. \tag{7}$$

The objective of $A_\theta(T)$ will be to learn the spatial variability of $A$ with respect to $T$ for multiple glaciers in different climates. In our toy model, this variability is described by prescribing an artificial law for which a reference dataset is generated. In order to generate this artificial law, we have used the relationship between ice temperature and $A$ from Cuffey and Paterson (2010), and replaced ice temperatures with a relationship between $A$ and $T$. This relationship is based on the hypothesis that $T$ is a proxy of $A$. However, it ignores many other important physical drivers influencing the value of $A$, such as the temperature of

ice, the type of fabric and the water content (Cuffey and Paterson, 2010). Nonetheless, this simple example serves to illustrate the modelling framework based on UDEs for glacier ice flow modelling, while acting as a platform to present both the technical challenges and adaptations performed in the process, and the future perspectives for applications at larger scales with additional data.

## 3   Experiment implementation in Julia

The combination of Python tools from OGGM with the UDE glacier modelling framework in Julia has resulted in the creation of a new Julia package named `ODINN.jl` (OGGM + DIfferential equation Neural Networks; Bolibar and Sapienza (2023)). For the purpose of this study, ODINN has been used to study the viability of UDEs to solve and learn subparts of the SIA equation. The use of `PyCall.jl` enables a seamless integration of Python libraries such as OGGM and `xarray` (Hoyer and Hamman, 2017) within ODINN.

**3.1   Training dataset**

The following data are used for the initial conditions of simulated glaciers: a Digital Elevation Model (DEM) for the glacier surface elevation $S$ based on the Shuttle Radar Topography Mission from the year 2005 (SRTM Farr et al. (2007)), and estimated glacier ice thickness $H$ from the global dataset from (Farinotti et al., 2019) based on the glacier outlines around the year 2003 of the Randolph Glacier Inventory (Consortium, 2017). All these datasets, together with all glacier information are

retrieved using the Open Global Glacier Model (OGGM), an open-source glacier evolution model in Python providing a vast set of tools to retrieve and process climate and topographical data related to glaciers (Maussion et al., 2019). Since these datasets are only available for just one or few timestamps, but have a global spatial coverage of almost all of the ∼220,000 glaciers on Earth, we perform this training for 17 different glaciers (see table in Appendix C) distributed in different climates around the

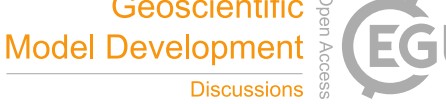

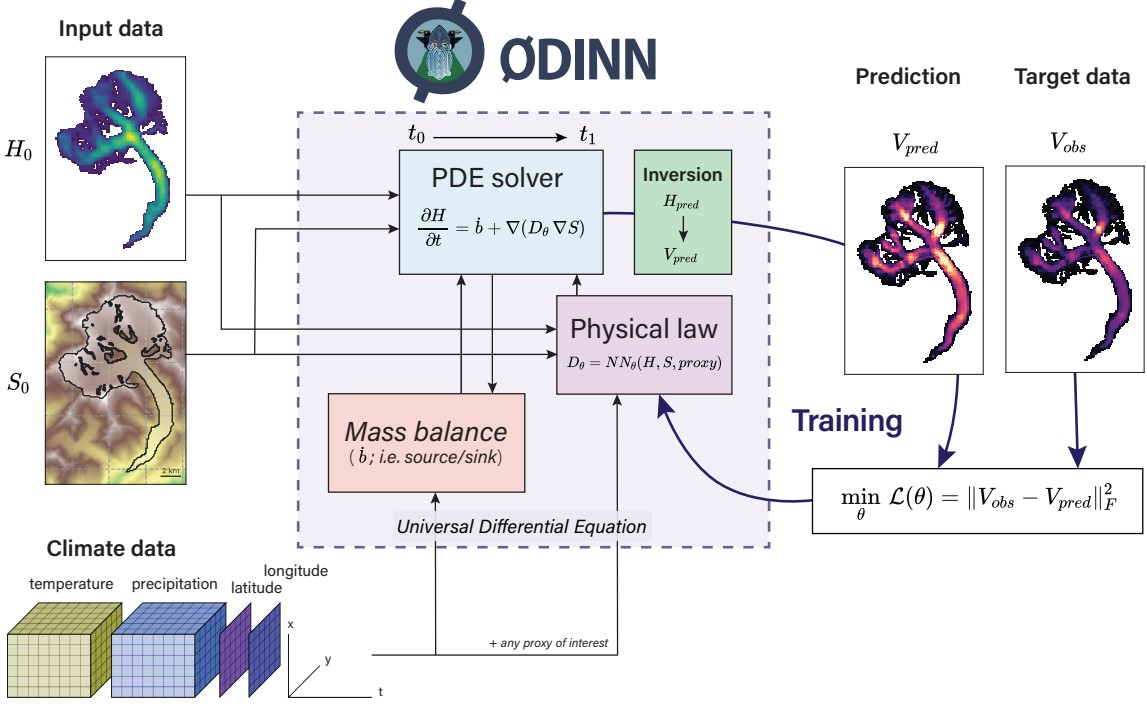

**Figure 2.** Overview of `ODINN.jl`'s workflow to perform functional inversions of glacier physical processes using Universal Differential Equations. The parameters ($\theta$) of a function determining a given physical process ($D_\theta$), expressed by a neural network $NN_\theta$, are optimized in order to minimize a loss function. For this study, the physical law was constrained only by climate data, but any other proxies of interest can be used to design it. The glacier mass balance is downscaled (i.e. it depends) on $S$, which is updated by the solver, thus dynamically updating the state of the simulation for a given timestep.

world. This enables a good sampling of different climate conditions from which to take $T$ to compute $A$. All climate data was
based on the W5E5 climate dataset (Lange, 2019), also retrieved using OGGM. For the purpose of the synthetic experiments,
some of the boundary conditions (surface topography, glacier bedrock inferred from topography and ice thickness) are assumed
to be perfectly known.

### 3.2 Differential equation solver

In order to solve the SIA Equation (2), we perform a discretization in the spatial domain to solve the problem as a combination
of Ordinary Differential Equations (ODEs). The solution of the ODEs is evaluated in a constant grid, determined by the
preprocessed datasets by OGGM. The resolution of the spatial grid is automatically adjusted depending on the glacier size and
domain size, typically ranging between 100x100 to 200x200 grid points, which leads to a system of coupled ODEs ranging
from 10,000 to 40,000 equations per glacier. All gradients and divergences are computed in a staggered grid to improve stability





(see Appendix A). Once the problem has been discretized in the spatial domain, we use a 5-stage, third order low-storage solver
with embedded error estimator (`RDPK3Sp35`) from `DifferentialEquations.jl` (Rackauckas and Nie, 2017) to solve
the SIA forward in time.

In order to create conditions similar to those one would encounter when using remote sensing observations for the functional
inversions, we add Gaussian noise with zero mean and standard deviation $6 \cdot 10^{-18}$ (around 30% of the average $A$ value) to the
output of the prescribed law (Figure 3). This setup is used to compute the reference solutions, which will then be used by the
UDE to attempt to infer the prescribed law indirectly from glacier ice surface velocities and/or ice thickness.

For the SIA UDE, we substitute $A$ by a small feed-forward neural network with a $(1, 3, 10, 3, 1)$ architecture, using `Flux.jl`
(Innes et al., 2018). Since the problem is highly constrained by the structure of the PDE, a very small neural network is enough
to learn the dynamics related to the subpart of the equation it is representing (i.e. $A$). This network has a single input and output
neuron, thus producing a one-to-one mapping. This small size is one of the main advantages of UDEs, which don't require as
much data as traditional data-driven machine learning approaches. We used a softplus activation function in all layers except for
the last layer, for which we use a rescaled sigmoid activation function which constrains the output within physically plausible
values (i.e. $8^{-20}$ to $8^{-17}$ $yr^{-1}Pa^{-3}$). The use of continuous activation functions has been proven to be more effective for
neural differential equations, since their derivatives are also smooth, thus avoiding problems of vanishing gradients (Kim et al.,
2021).

## 3.3 Surface mass balance

In order to compute the glacier surface mass balance (i.e. a source/sink) for both the PDEs and the UDEs, we used a very simple
temperature-index model with a single melt factor and a precipitation factor set to 5 mm $d^{-1}\,^{\circ}C^{-1}$ and 1.2, respectively. These
are average values found in the literature (Hock, 2003), and despite its simplicity, this approach serves to add a realistic MB
signal on top of the ice rheology in order to assess the performance of the inversion method under realistic conditions. Instead of
adding the mass balance term $\dot{b}$ in the SIA Equation (2) directly to the updates of the steps in the numerical solver, it was added
to the SIA using a `DiscreteCallback` from `DifferentialEquations.jl`. This enabled the correct modification of
the $H$ matrix with any desired time intervals and without producing numerical instabilities (Rackauckas and Nie, 2017). We
observed this makes the solution to be more stable without the need of reducing the stepsize of the solver.

In order to find a good compromise between computational efficiency and memory usage, we pre-process raw climate files
from W5E5 (Lange, 2019) for the simulation period of each glacier. Then, within the run, for each desired timestep where
the surface mass balance needs to be computed (monthly by default), we read the raw climate file for a given glacier and
we downscale the air temperature to the current surface elevation $S$ of the glacier given by the SIA PDE. For that, we use
the adaptive lapse rates given by W5E5, thus capturing the topographical feedback of retreating glaciers in the surface mass
balance signal (Bolibar et al., 2022).



## 3.4 Sensitivity methods and differentiation

In order to minimize the loss function from Equation (6), we need to evaluate its gradient with respect to the parameters of the neural network and then perform gradient-based optimization. Different methods exist to evaluate the sensitivity or gradients of the solution of a differential equation. These can be classified depending if they run backwards or forward with respect to the solver. Furthermore, they can be divided between continuous and discrete, depending on if they manipulate a discrete evaluation of the solver or if instead they solve an alternative system of differential equations in order to find the gradients (see Figure B1). For a more detailed overview on this topic, see Appendix B.

Here we compare the evaluation of the gradients using a continuous adjoint method integrated with automatic differentiation and a hybrid method that combines automatic differentiation with finite differences.

### 3.4.1 Continuous adjoint sensitivity analysis

For the first method based on pure automatic differentiation, we used the `SciMLSensitivity.jl` package, an evolution of the former `DiffEqFlux.jl` Julia package (Rackauckas et al., 2019), capable of weaving neural networks and differential equation solvers with AD. In order to train the SIA UDE from Equation (7), we use the same previously mentioned numerical scheme as for the PDE (i.e. `RDPK3Sp35`). Unlike in the original Neural ODEs paper Chen et al. (2019), simply reversing the ODE for the backward solve results in unstable gradients. This has been shown to be the case for most differential equations, particularly stiff ones like the one from this study (Kim et al., 2021). In order to overcome this issue, we used the full solution of the forward pass to ensure stability during the backward solve, using the `InterpolatingAdjoint` method as described in the Stiff Neural ODEs paper (Kim et al., 2021). This method combined with an explicit solver proved more efficient than using a quadrature adjoint. It is also possible to use checkpointing to interpolate between fewer stored data points. This has the advantage of reducing memory usage at the cost of sacrificing some computational performance. To compute the vector-Jacobian products involved in the adjoint calculations of the SIA UDE, we used reverse-mode AD with the `ReverseDiff.jl` package with a cached compilation of the reverse tape. We found that for our problem, the limitation of not being able to use control flow was easily bypassed, and performance was noticeably faster than other AD packages in Julia, such as `Zygote.jl`.

### 3.4.2 Finite differences

The second method consists in using AD just for the neural network and finite differences for capturing the variability of the loss function with respect to the parameter $A$. Notice that in Equation (6) we can write $\mathcal{L}_k(\theta) = \mathcal{L}_k(A_\theta(T_k))$, with $A(\theta)$ the function that maps input parameters $T_k$ into the scalar value of $A$ (which for this example is assumed to be a single scalar across the glacier) as a function of the neural network parameters $\theta$. Once $A$ has being specified, the function $\mathcal{L}_k$ is a one-to-one function that is easily differentiable using finite differences. If we define $\nabla_\theta A(T_k)$ the gradient of the neural network that we obtain using AD, then the full gradient of the loss function can be evaluated using the centered numerical approximation

$$\nabla_\theta \mathcal{L}_k \approx \frac{\mathcal{L}(\theta + \eta \nabla_\theta A(T_k)) - \mathcal{L}(\theta - \eta \nabla_\theta A(T_k))}{2\eta \|\nabla_\theta(T_k)\|^2} \nabla_\theta A(T_k), \tag{8}$$





where $\eta$ is the stepsize used for the numerical approximation of the derivative. Notice that the first term on the right hand side is just a scalar that quantifies the sign and amplitude of the gradient, which will be always in the direction of $\nabla_\theta A(T_k)$. The choice of stepsize $\eta$ is critical in order to correctly estimate the gradient. Notice that this method works just when there are a few parameters, and will not generalize well to the case of an $A$ that varies in space and time for each glacier. The main

advantage of this method is that it runs orders of magnitude faster than a pure AD one.

### 3.5 Optimization

Once the gradient has been computed by one of the previous methods, optimization of the total loss function without any extra regularization penalty to the weights in the loss function was performed using a Broyden–Fletcher–Goldfarb–Shanno (BFGS) optimizer with parameter $0.001$. We also tested ADAM (Mogensen and Riseth, 2018) with a learning rate of $0.01$. BFGS

converges in fewer epochs than ADAM, but it had a higher computational cost per epoch. Overall, BFGS performed better in different scenarios, resulting in a more reliable UDE training. For this toy model, a full epoch was trained in parallel using 17 workers, each one for a single glacier. Larger simulations will require batching either across glaciers, or across time in case a dataset with dense temporal series were used.

### 3.6 Scientific computing in the cloud

In recent years, scientific workflows in Earth and environmental sciences have benefited from transitioning from local to cloud computing (Gentemann et al., 2021; Abernathey et al., 2021; Mesnard and Barba, 2020). The benefits of cloud computing are two-fold. On one side, it facilitates the creation of shared computational environments, datasets and analysis pipelines, which leads to more reproducible scientific results by enabling standalone software containers (e.g. Binder) for other researchers to easily reproduce scientific results. On the other hand, working in cloud environments lowers the access bar for many scientists

from unrepresented groups across multiple institutions and continents.

As part of this new approach in terms of geoscientific computing, we are computing everything directly in the cloud using a JupyterHub through the 2i2c infrastructure. This JupyterHub allows us to work with both Julia and Python, using Unix terminals, Jupyter notebooks (Thomas et al., 2016) and VSCode directly on the browser. Moreover, this provides a lot of flexibility in terms of computational resources. When logging in, one can choose between different machines, ranging from

small ones (1-4 CPUs, 4-16 GB RAM) to very large ones (64 CPUs, 1 T4 Tensor Core GPU, 1 TB RAM), depending on the task to be run. The largest simulations for this study were run in a Large machine, with 16 CPUs and 64 GB of RAM.

## 4 Results

Despite its apparent simplicity, it is not a straightforward problem to invert the spatial function of $A$ with respect a predictor indirectly from surface velocities, mainly due to the highly nonlinear behaviour of the diffusivity of ice (see Equation (2)). We

ran a functional inversion using two different differentiation methods for 17 different glaciers (see Table C1) for a period of 5 years.



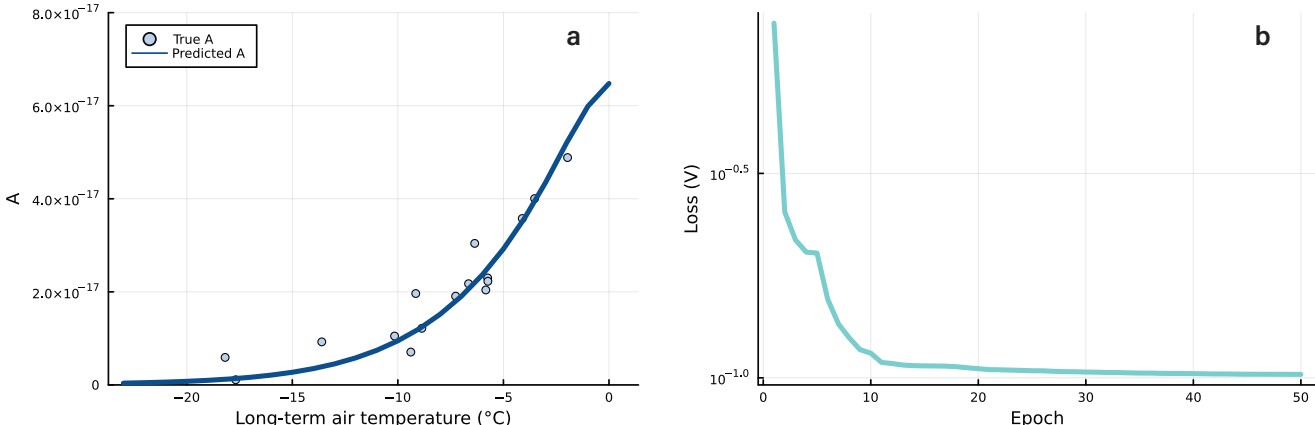

**Figure 3.** (a) Inferred function by the NN embedded in the SIA PDE using full automatic differentiation. The NN learnt the prescribed noisy function (each dot represents a glacier), that relates Glen's coefficient $A$ with a proxy of interest (i.e. the long-term air temperature $T$). (b) Evolution of the loss through training, using a BFGS optimizer. The loss is based on a scaled mean squared error of the difference between the simulated and target ice surface velocities. The scaling is used to correctly account for values of different orders of magnitude. Without any use of regularization, the optimization converges in around 20 epochs. Note that the loss is shown in log scale.

Training the UDE with full batches using the continuous adjoint method described in Appendix B6 converges in around 20 epochs. The NN is capable of successfully approximating the prescribed nonlinear function of $A$. The loss sees a steep decrease in the first epochs, with BFGS optimizing the function towards the lowest values of $A$, which correctly approximate

the majority of values of the prescribed nonlinear function. From this point, the function slowly converges until it finds an optimal non-overfitted solution (Figure 3b). This simulation took about 3h to converge, with a running time between 6 to 12 minutes per epoch, in a machine in the cloud-based JupyterHub with 16 CPUs and 64 GB of RAM, using all available CPUs to simulate in parallel the 17 glaciers in batches and using the full 64 GB of memory. Figure 3a shows the parametrization of $A$ as a function of $T$ obtained with the trained NN. We observe that the neural network is able to capture the monotonic increasing

function $A(T)$ without overfitting the noisy values used for training (dots in the plot). Interestingly, the lack of regularization did not affect overfitting. We are unsure about the reasons behind this behaviour, but we suspect this could be related to an implicit regularization caused by UDEs. This property has not been studied yet, so more investigations should be carried out in order to better understand this apparent robustness to overfitting.

We also compared the efficiency of our approach when using the finite differences scheme. Since this does not require heavy

backwards operations as the continuous adjoint method does, the finite difference method runs faster (around 1 minute per epoch). However, we encountered difficulties in picking the right stepsize $\eta$ in Equation (8). Small values of $\eta$ lead to floating number arithmetic errors and large $\eta$ to biased estimates of the gradient. On top of this effect, we found that numerical precision errors in the solution of the differential equation result in wrong gradients and the consequent failure in the optimization procedure (see discussion about this in Section 5.2.3). A solution for this would be to pick $\eta$ adaptive as we update the value





of the parameter $\theta$. However, this would lead to more evaluations of the loss function. Instead, we applied early stopping when we observed that the loss function reached a reasonable minimum.

### 4.1 Robustness to noise in observations

The addition of surface mass balance (i.e. a source/sink) to the SIA equation further complicates things for the functional inversion, particularly from a computational point of view. The accumulation and ablation (subtraction) of mass on the glacier

introduces additional noise to the pure ice flow signal. The mass loss in the lower elevations of the glacier slows down ice flow near the tongue, whereas the accumulation of mass in the higher elevations accelerates the ice flow on the upper parts of the glacier.

As an experiment to test the robustness of the functional inversions made by the UDE, we used different surface mass balance models for the reference simulation (i.e. the ground truth), and the UDE. This means that the surface mass balance

signal during training is totally different from the one in the ground truth. We achieved this by using a temperature-index model with a melt factor of 4 mm $d^{-1}$ ° $C^{-1}$ and a precipitation factor of 2 for the reference run, and a melt factor of 8 mm $d^{-1}$ ° $C^{-1}$ and a precipitation factor of 1 for the UDE. This means that the UDE is perceiving a much more negative surface mass balance than the actual one from the ground truth. Despite the really large difference that can be seen in Figure 4, the UDE was perfectly capable of inverting the nonlinear function of $A$. The evolution of the loss was less smooth than for the

case of matching surface mass balance rates, but it also converged in around 20 epochs, with no noticeable different in final performance.

This shows the robustness of this modelling approach, particularly when the ice surface velocities ($V$) are used in the loss function. Unlike the glacier ice thickness, $V$ is much less sensitive to changes in surface elevation, making it a perfect data for inferring ice rheology properties. This is also due to the fact that we are using ice surface velocities averaged across multiple

years, which lessen the importance of changes in surface elevation. When working with velocities with a higher temporal resolution, these will likely become more sensitive to noise. This weak dependence on the surface mass balance signal will be highly beneficial for many applications, since it will imply that the inversion can be done even if we only have an approximate representation of the surface mass balance, which will be the case for many glaciers.

## 5 Discussion: challenges and perspectives

### 5.1 Application to functional inversions of glacier physical processes

This first implementation of a UDE on glacier ice flow modelling serves as a baseline to tackle more complex problems with large datasets. One main simplification of this current setup needs to be overcome in order to make the model useful at a global scale for exploring and discovering empirical laws. In this study, only ice deformation (creep) has been taken into account in the diffusivity. Basal sliding, at the ice-bedrock interface, will have to be included in the SIA equation to accommodate different

configurations and behaviours of many glaciers around the world. Therefore, a logical next step would be to infer $D$ in Equation

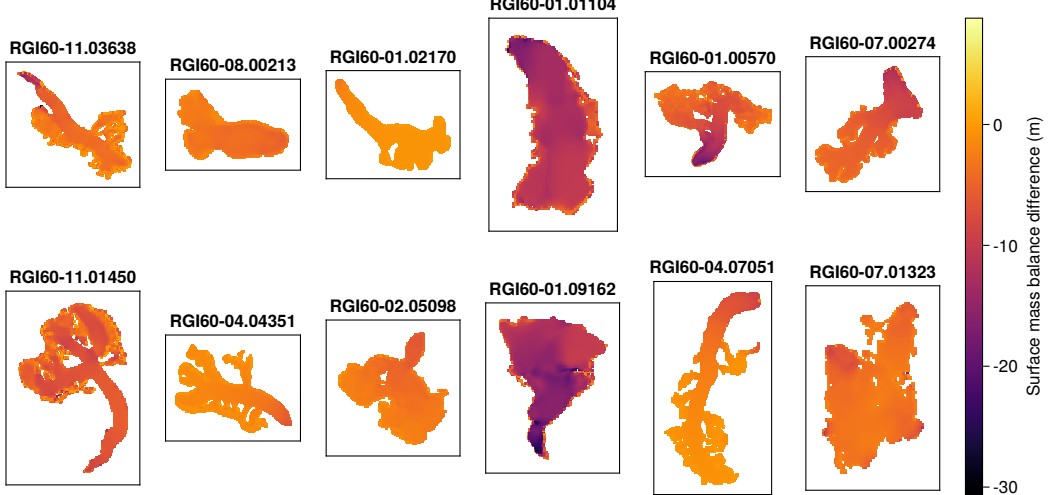

**Figure 4.** Differences in surface elevation for a 5-year simulation, coming from the different applied mass balance rates, between the ground truth data and the training of the UDE. Despite the "noise" coming from the different mass balance signal, the UDE is perfectly capable of learning the underlying nonlinear function of $A$. This proves the robustness against noise of this functional inversion framework for glacier rheology when using ice surface velocities. Showing only 12 out of the total 17 glaciers.

(3), including the sliding coefficient $C$ from Equation (2) using a UDE. Nonetheless, this is by nature an ill-posed problem, since the only available ice velocity observations are from the surface, encompassing both creep and basal sliding. This results in degeneracy (i.e. equifinality), making it very challenging to disentangle the contributions of each physical processes to ice flow. This is particularly true for datasets with average annual surface velocities, since both physical processes contribute

to glacier velocities, with no obvious way of separating them. In order to overcome such issues, using new remote sensing datasets with high temporal resolution, like Nanni et al. (2023) with observations every 10 days, can help better constrain the contribution of each physical process. This implies that we cannot only exploit the spatial dimension with multiple glaciers, but also rich time series of the fluctuations of glacier velocities along the seasons. Such dynamics can help disentangle the main contributions of creep during the winter season, and the onset of sliding during the summer season as the subglacial

hydrological network activates due to melt.

Interestingly, depending on the used ice surface velocity observations, the need of a numerical solver and a UDE are not imperative for a functional inversion. For a single snapshot of ice surface velocities between two dates (e.g. 2017-2018 in Millan et al. (2022)), a functional inversion can be performed directly on the SIA equation without the need of a solver. The average ice surface velocities can be direcly inverted if the initial conditions are known. This reduces the technical complexity enormously,

enabling one to focus on more complex NN architectures and functions to inverse ice rheology and basal sliding properties. Some initial tests have shown that such problems train orders of magnitude faster. However, since only one timestamp is present





for the inversions, the inversion is extremely sensitive to time discrepancies in the input datasets, making it potentially quite vulnerable to noisy or temporally misaligned datasets.

Alternatively, the optimization of the NN for ice rheology inference based on ice surface velocities has proved to be robust to the noise added by the SMB component. This serves to validate an alternative glacier ice dynamics model calibration strategy to those of the majority of large-scale glacier models (e.g. OGGM and GloGEM; Huss and Hock (2015)). By being able to calibrate separately ice rheology and MB, one can avoid many equifinality problems that appear when attempting to calibrate both components at the same time. A classic problem of a joint calibration is the ambiguity in increasing/decreasing accumulation vs increasing/decreasing Glen's coefficient ($A$). `ODINN.jl`, with its fully differentiable codebase, provides a different strategy consisting in two main steps: (i) Calibrating the ice rheology from observed ice surface velocities (Millan et al., 2022), observed ice thicknesses (Consortium, 2019) and DEMs; (ii) calibrating the MB component (e.g. melt and accumulation factors) with the previously calibrated ice rheology based on both point glaciological MB observations and multiannual geodetic MB observations (Hugonnet et al., 2020). This maximises the use of current glaciological datasets for model calibration, even for transient simulations. Such a differentiable modelling framework presents both the advantages of complex inversions and the possibility of performing data assimilation with heterogeneous and sparse observations.

## 5.2 Scientific machine learning

### 5.2.1 Automatic differentiation approaches

Automatic differentiation is a centerpiece of the modelling framework presented in this study. In the Julia programming language, multiple AD packages exist, which are compatible with both differential equation and neural networks packages, as part of the SciML ecosystem. Julia source code is automatically differentiable, enabling the user to switch between different AD backends with very few or no changes to the source code. Each package has advantages and disadvantages, which make them suitable for different tasks. In our case, `ReverseDiff.jl` turned out to be the best performing AD package, due to the speed gained by reverse tape compilation. Together with `Zygote.jl` (Innes et al., 2019), another popular backwards AD package, they have the limitation of not allowing mutation of arrays. This implies that no in-place operations can be used, thus augmenting the memory allocation of the simulations considerably. `Enzyme.jl` (Moses and Churavy, 2020) is arising as a promising alternative, with the fastest gradient computation times in Julia (Ma et al., 2021). It directly computes gradients on statically analyzable LLVM, without differentiating any Julia source code. Nonetheless, `Enzyme.jl` is still under heavy development, and it is still not stable enough for problems like the ones from this study. As `Enzyme.jl` will become more robust in the future, appears likely to become the *de facto* solution for AD in Julia.

Overall, the vision on AD from Julia is highly ambitious, attempting to perform AD directly on source code, with minimal impact on the user side and with the possibility of easily switching AD back-ends. In practice, this implies that it is much more complex to achieve from a technical point of view than hardcoded gradients linked to specific operators, an approach followed by JAX (Bradbury et al., 2020) and other popular deep learning Python libraries. On the short term, the latter provides a more stable experience, albeit a more rigid one. However, in the long term, once these packages are given the time to



The dependency of the error term $\text{Err}(\theta, \text{hyper}(\theta))$ as a function of $\theta$ induce wiggles in the loss function. This is a problem that was observed using finite differences to differentiate part of the loss function (see Section 3.4). The gradient calculated by making variation of the parameter $\theta$ capture the variations of $\text{Err}(\theta, \text{hyper}(\theta))$, which lead to spurious gradients. On the other side, automatic differentiation compute gradients using one single evaluation of $\theta$, meaning that it differentiates only the term $\text{Solver}(u_0, t_0, t_1, \theta)$, being robust to the error term. Further investigation is needed in order to establish the effect of these wiggles during optimization and how these can distort the gradient obtain using different sensitivity methods.

Another interesting question regards the training and regularization of UDEs and related physics-informed neural networks. During training, we observed that the neural network never overfitted the noisy version of prescribed law $A(T)$ (see Section 3.2). We conjecture that one reason why this may be happening is because of the implicit regularization imposed by the form of the differential equation in Equation (2).

## 6 Conclusions

Despite the ever increasing amounts of new Earth observations coming from remote sensing, it is still extremely challenging to translate complex, sparse, noisy data into actual models and physical laws. Paraphrasing Rackauckas et al. (2020), "In the context of science, the well-known adage *a picture is worth a thousand words* might well be *a model is worth a thousand datasets*". Therefore, there is a clear need for new modelling frameworks capable of generating data-driven models with the interpretability and constraints of classic physical models. Universal Differential Equations (UDEs) embed a universal function approximator (e.g. a neural network) inside a differential equation. This enables additional flexibility typical from data-driven models into a reliable physical structure determined by a differential equation.

We presented `ODINN.jl`, a new modelling framework based on UDEs applied to glacier ice flow modelling. We illustrated how UDEs, supported by differentiable programming in the Julia programming language, can be used to retrieve empirical laws present in datasets, even in the presence of noise. We did so by using the Shallow Ice Approximation PDE, and learning a prescribed artificial law as a subpart of the equation. We used a neural network to infer Glen's coefficient $A$, determining the ice viscosity, with respect to a climatic proxy for 17 different glaciers across the world. The presented functional inversion framework is robust to noise present in input observations, particularly on the surface mass balance, as shown in an experiment.

This study can serve as a baseline for other researchers interested in applying UDEs to similar nonlinear diffusivity problems. It also provides a codebase to be used as a backbone to explore new parametrizations for large-scale glacier modelling, such as for glacier ice rheology, basal sliding, or more complex hybrid surface mass balance models.

*Code availability.* The source code of ODINN.jl v0.2.0 (Bolibar and Sapienza, 2023) used in this study is available as an open-source Julia package: https://github.com/ODINN-SciML/ODINN.jl. The package includes Continuous Integration tests, installation guidelines on how to use the model and a Zenodo DOI: https://zenodo.org/record/8033313. OGGM v1.6 (Maussion et al., 2023) is also available as an open-source Python package at: https://github.com/OGGM/oggm, with documentation and tutorials available at https://oggm.org.





## Appendix A: Numerical Solution

Solving Equation (2) is challenging since the diffusivity $D$ depends on the solution $H$. Furthermore, the SIA equation is a
*degenerate diffusion equation* since the diffusivity $D$ vanishes as $H \to 0$ or $\nabla S \to 0$. In cases where the glacier does not
extend to the margins of our domain, this allow us to treat the SIA equation as a free-boundary problem. However, at the
moment of solving the differential equation with a numerical solver, we still need to impose the constraint the the ice thickness
is non-negative, that is, $H \geq 0$.

We are going to consider a uniform grid on points $(x_j, y_k)$, with $j = 1, 2, \ldots, N_x$ and $k = 1, 2, \ldots, N_y$, with $\Delta x = \Delta y =
x_{j+1} - x_j = y_{k+1} - y_k$. Starting from some initial time $t_0$, we are going to update the value of the solution for $H$ by steps $\Delta t_i$,
with $t_i = t_{i-1} + \Delta t_{i-1}$. We are going to refer as $H_{j,k}^i$ for the numerical approximation of $H(t_i, x_j, y_k)$. In this way, we are
going to have a system of ODEs for each $H_{j,k}$.

An important consideration when working with numerical schemes for differential equations is the stability of the method.
Here we are going to consider just explicit methods, although the spatial discretization is the same for implicit methods. Explicit
methods for the SIA equation are conditionally stable. In order to get stability, we need:

1. To evaluate the diffusivity in a staggered grid $D_{i\pm\frac{1}{2}, k\pm\frac{1}{2}}$ labeled by semi-integers indices (circles on the dotted grid in
   Figure A1).

2. To choose a temporal step size $\Delta t$ such that $\Delta t \leq \Delta x^2 / 4 D_{\max}$, where $D_{\max}$ is the maximum diffusivity.

The algorithm to solve the SIA equation follows the next steps.

1. Assume we know the value of $H_{j,k} = H_{j,k}^i$ at some give time $t_i$. Then, we are going to compute all the diffusivities on
   the staggered grid. As we mentioned before, the diffusivity is a function of $H$, $\frac{\partial S}{\partial x}$ and $\frac{\partial S}{\partial y}$. Instead of using one single
   estimate of $H$ to approximate all these quantities, the idea is to use averages quantities on the primal grid to compute the
   diffusivity on the staggered grid (red arrows in Figure A1). We define the averages quantities

$$H_{j+\frac{1}{2}, k+\frac{1}{2}} = \frac{1}{4} \left( H_{j,k} + H_{j+1,k} + H_{j,k+1} + H_{j+1,k+1} \right) \tag{A1}$$

$$\left( \frac{\partial S}{\partial x} \right)_{j+\frac{1}{2}, k+\frac{1}{2}} = \frac{1}{2} \left( \frac{H_{j+1,k} - Hj,k}{\Delta x} + \frac{H_{j+1,k+1} - H_{j,k+1}}{\Delta x} \right) \tag{A2}$$

$$\left( \frac{\partial S}{\partial y} \right)_{j+\frac{1}{2}, k+\frac{1}{2}} = \frac{1}{2} \left( \frac{H_{i,k+1} - H_{i,k}}{\Delta y} + \frac{H_{i+1,k+1} - H_{i+1,k}}{\Delta y} \right). \tag{A3}$$

Then, we compute the diffusivity on the staggered grid as

$$D_{j+\frac{1}{2}, k+\frac{1}{2}} = D \left( H_{j+\frac{1}{2}, k+\frac{1}{2}}, \left( \frac{\partial S}{\partial x} \right)_{j+\frac{1}{2}, k+\frac{1}{2}}, \left( \frac{\partial S}{\partial y} \right)_{j+\frac{1}{2}, k+\frac{1}{2}} \right). \tag{A4}$$





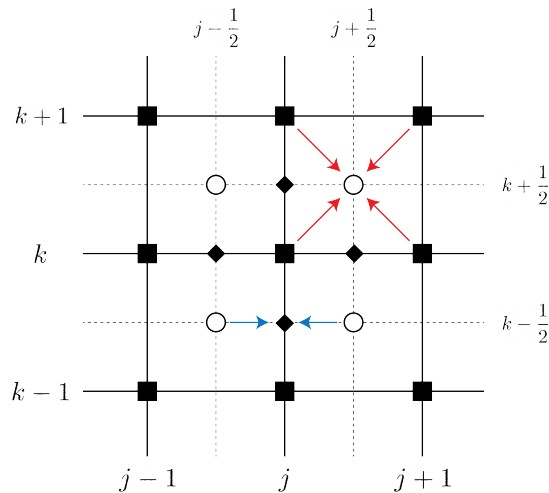

**Figure A1.** Staggered grid, used to solve the Shallow Ice Approximation PDE. Black squares represent the primal grid, empty circles the staggered grid, blue arrows operations in the edges of the primal grid, and red arrows operations in the staggered grid.

2. Compute (another) average diffusivity but now on the edges of the primal grid (blue arrows in Figure A1):

$$D_{j,k\pm\frac{1}{2}} = \frac{1}{2}\left(D_{j-\frac{1}{2},k\pm\frac{1}{2}} + D_{j+\frac{1}{2},k\pm\frac{1}{2}}\right) \tag{A5}$$

$$D_{j\pm\frac{1}{2},k} = \frac{1}{2}\left(D_{j\pm\frac{1}{2},k-\frac{1}{2}} + D_{j\pm\frac{1}{2},k+\frac{1}{2}}\right) \tag{A6}$$

3. Compute the diffusive part of the SIA equations on the point in the primal grid $(j,k)$ as

$$\nabla(D\nabla S)_{j,k} = \frac{D_{j+\frac{1}{2},k}(S_{j+1,k} - S_{j,k}) - D_{j-\frac{1}{2},k}(S_{j,k} - S_{j-1,k})}{\Delta x^2}$$
$$+ \frac{D_{j,k+\frac{1}{2}}(S_{j,k+1} - S_{j,k}) - D_{j,k-\frac{1}{2}}(S_{j,k} - S_{j,k-1})}{\Delta y^2} \tag{A7}$$

4. Update the value of $H$ following an explicit scheme. Here we use the different solvers available in DifferentialEquations.jl. Just for illustration, a simple explicit scheme could be one that updates the values of the ice thickness following


$$H_{j,k}^{i+1} = H_{j,k}^i + \Delta t_i \left( \dot{b}_{j,k}^i + \nabla(D\nabla S)_{j,k}^i \right). \tag{A8}$$

where $\Delta t_i$ is the time step, which in our case is automatically selected by the numerical solver to ensure stability.

In practice, the stepsize $\Delta t_i$ is chosen automatically by the numerical solver included in `DifferentialEquations.jl` (Rackauckas and Nie, 2017). However, this does not automatically guarantees that the updates in the ice thickness (Equation





(A8)) make $H_{j,k}^{i+1} \geq 0$. A sufficient condition is given by

$$-H_{j,k}^i \leq S_{j\pm1,k} - S_{j,k}, \tag{A9}$$

$$-H_{j,k}^i \leq S_{j,k\pm1} - S_{j,k}. \tag{A10}$$

This condition guarantees that the computed diffusivity (Equation (A7)) satisfies

$$\nabla(D\nabla S)_{j,k}^i \geq -\frac{4D_{\max}}{\Delta x^2}H_{j,k}^i \tag{A11}$$

and hence

$$H_{j,k}^{i+1} \geq H_{j,k}^i + \Delta t_i \dot{b}_{j,k}^i - \frac{4\Delta t_i D_{\max}}{\Delta x^2}H_{j,k}^i = \left(1 - \frac{4\Delta t_i D_{\max}}{\Delta x^2}\right)H_{j,k}^i + \Delta t_i \dot{b}_{j,k}^i \geq \Delta t_i \dot{b}_{j,k}^i, \tag{A12}$$

where the last inequality is consequence of the stability condition $\Delta t_i \leq \Delta x^2/4D_{\max}$. In cases where no mass balance term is added ($\dot{b}_{j,k}^i = 0$), we simply have that $H_{j,k}^{i+1} \geq 0$ for all grid points. In the general case with mass balance, we still need to clip the updated ice thickness by replacing $H_{j,k}^{i+1}$ by $\max\{0, H_{j,k}^{i+1}\}$. This include those cases with excessive ablation.

**Appendix B: Overview of sensitivity methods**

In this section we provide a high-level explanation of the two methods we used compute gradient of functions involving solutions of differential equations, namely finite differences and continuous adjoint sensitivity analysis. Consider a system of ordinary differential equations given by

$$\frac{du}{dt} = f(u,\theta,t), \tag{B1}$$

where $u \in \mathbb{R}^n$ and $\theta \in \mathbb{R}^p$. We are interested in computing the gradient of a given loss function $L(u(\cdot,\theta))$ with respect to the parameter $\theta$.

**B1   Finite differences**

The simplest way of evaluating a derivative is by computing the difference between the evaluation of the function at a given point and a small perturbation of the function. In the case of a loss function, we can approximate


$$\frac{dL}{d\theta_i}(\theta) \approx \frac{L(\theta + \varepsilon e_i) - L(\theta)}{\varepsilon}, \tag{B2}$$

with $e_i$ the $i$-th canonical vector and $\varepsilon$ a small number. Even better, it is easy to see that the centered difference scheme

$$\frac{dL}{d\theta_i}(\theta) \approx \frac{L(\theta + \varepsilon e_i) - L(\theta - \varepsilon e_i)}{2\varepsilon}, \tag{B3}$$

leads also to more precise estimation of the derivative.

However, there are a series of problems associated to this approach. The first one is how this scales with the number of
parameters $p$. Each directional derivative requires the evaluation of the function twice. For the centered differences approach





in Equation (B3), this will require $2p$ function evaluations, which demand to solve the differential equation in forward mode each time. A second problem is due to truncation errors. Equation (B2) involves the subtraction of two numbers that are very close to each other. As $\varepsilon$ gets smaller, this will lead to truncation of the subtraction, introducing numerical errors than then will be amplified by the division by $\varepsilon$. Due to this, some heuristics have been introduced in order to pick the value of $\varepsilon$ that will

minimize the error, specifically picking $\varepsilon^* = \sqrt{\varepsilon_{\text{machine}}} \|\theta\|$, with $\varepsilon_{\text{machine}}$ the machine precision (e.g. `Float64`). However, in practice it is difficult to pick a value of $\epsilon$ that leads to universal good estimations of the gradient.

### B2    Continous Adjoint Sensitivity Analysis (CASA)

Consider an integrated loss function of the form

$$L(u;\theta) = \int_{t_0}^{t_1} h(u(t;\theta),\theta)dt, \tag{B4}$$

which includes the simple case of the loss function $\mathcal{L}_k(\theta)$ in Equation (6). Using the Lagrange multiplier trick, we can write a new loss function $I(\theta)$ identical to $L(\theta)$ as

$$I(\theta) = L(\theta) - \int_{t_0}^{t_1} \lambda(t)^T \left( \frac{du}{dt} - f(u,t,\theta) \right) dt \tag{B5}$$

where $\lambda(t) \in \mathbb{R}^n$ is the Lagrange multiplier of the continuous constraint defined by the differential equation. Now,

$$\frac{dL}{d\theta} = \frac{dI}{d\theta} = \int_{t_0}^{t_1} \left( \frac{\partial h}{\partial \theta} + \frac{\partial h}{\partial u}\frac{\partial u}{\partial \theta} \right) dt - \int_{t_0}^{t_1} \lambda(t)^T \left( \frac{d}{dt}\frac{du}{d\theta} - \frac{\partial f}{\partial u}\frac{du}{d\theta} - \frac{\partial f}{\partial \theta} \right) dt. \tag{B6}$$

We can derive an easier expression for the last term using integration by parts. We define the sensitivity as $s = \frac{du}{d\theta}$, and then perform integration by parts in the term $\lambda^T \frac{d}{dt}\frac{du}{d\theta}$ we derive

$$\frac{dL}{d\theta} = \int_{t_0}^{t_1} \left( \frac{\partial h}{\partial \theta} + \lambda^T \frac{\partial f}{\partial \theta} \right) dt - \int_{t_0}^{t_1} \left( -\frac{d\lambda^T}{dt} - \lambda^T \frac{\partial f}{\partial u} - \frac{\partial h}{\partial u} \right) s(t)\,dt - \left( \lambda(t_1)^T s(t_1) - \lambda(t_0)^T s(t_0) \right). \tag{B7}$$

Now, we can force some of the terms in the last equation to be zero by solving the following adjoint differential equation for $\lambda(t)^T$ in backwards mode

$$\frac{d\lambda}{d\theta} = -\left( \frac{\partial f}{\partial u} \right)^T \lambda - \left( \frac{\partial h}{\partial u} \right)^T, \tag{B8}$$

with final condition $\lambda(t_1) = 0$. Then, in order to compute the full gradient $\frac{dL}{d\theta}$ we do

1. Solve the original differential equation $\frac{du}{dt} = f(u,t,\theta)$;

2. Solve the backwards adjoint differential equation (B8);





3. Compute the simplified version of the full gradient in Equation (B7) as

$$\frac{dL}{d\theta} = \lambda^T(t_0)s(t_0) + \int_{t_0}^{t_1}\left(\frac{\partial h}{\partial \theta} + \lambda^T\frac{\partial f}{\partial \theta}\right)dt. \tag{B9}$$

In order to solve the adjoint equation, we need to know $u(t)$ at any given time. There are different ways in which we can accomplish this: i) we can solve for $u(t)$ again backwards; ii) we can store $u(t)$ in memory during the forward step; or iii) we can do checkpointing to save some reference values in memory and interpolate in between. Computing the ODE backwards can be unstable and lead to exponential errors, (Kim et al., 2021). In Chen et al. (2019), the solution is recalculated backwards together with the adjoint simulating an augmented dynamics. One way of solving this system of equations that ensures stability is by using implicit methods. However, this requires cubic time in the total number of ordinary differential equations, leading to a total complexity of $\mathcal{O}((n+p)^3)$ for the adjoint method. Two alternatives are proposed in Kim et al. (2021), the first one called *Quadrature Adjoint* produces a high order interpolation of the solution $u(t)$ as we move forward, then solve for $\lambda$ backwards using an implicit solver and finally integrating $\frac{dL}{d\theta}$ in a forward step. This reduces the complexity to $\mathcal{O}(n^3+p)$, where the cubic cost in the number of ODEs comes from the fact that we still need to solve the original stiff differential equation in the forward step. A second but similar approach is to use a implicit-explicit (IMEX) solver, where we use the implicit part for the original equation and the explicit for the adjoint. This method also will have complexity $\mathcal{O}(n^3+p)$.

## Appendix C: Glaciers used in the study

Table C1 includes all the details of the glaciers used in this study to train the UDE. Glaciers were picked randomly across the world to sample different climates with long-term air temperatures ranging from -20º C to close to 0º C. This data was retrieved using OGGM's preprocessing tools from the Randolph Glacier Inventory v6 (Consortium, 2017). Note that OGGM processes the necessary gridded data (i.e. DEMs, ice thickness data) in a constant adaptive grid, which depends on glacier size.

*Author contributions.* J.B. conceived the project, designed the study, developed the model, wrote the paper and made the figures. F.S. designed the study, developed the model, investigated the sensitivity methods and wrote the paper. F.M. helped retrieving the datasets to force the model with OGGM, contributed to the coupling between both models and provided glaciological advice. R.L. helped with the experiment design and technical choices. B.W. provided glaciological advice and helped design the project. F.P. provided advice on the methods, software development and helped design the project. All authors contributed to the writing of the manuscript by providing comments and feedback.

*Competing interests.* At least one of the (co-)authors is a member of the editorial board of Geoscientific Model Development.



| RGI ID | Glacier name | Region | Area ($km^2$) | Lon/Lat (°) | Grid size | Grid res (m) |
|---|---|---|---|---|---|---|
| RGI60-11.03638 | Glacier d'Argentière | Central Europe | 13.79 | (6.98, 45.95) | (138, 129) | 62 |
| RGI60-11.01450 | Aletschgletscher | Central Europe | 82.2 | (8.02, 46.50) | (107, 154) | 137 |
| RGI60-08.00213 | Storglaciären | Scandinavia | 3.40 | (18.57, 67.90) | (110, 75) | 36 |
| RGI60-04.04351 | - | Arctic Canada South | 24.77 | (-63.21, 66.52) | (132, 104) | 80 |
| RGI60-01.02170 | Esetuk Glacier | Alaska | 7.5 | (-144.30, 69.29) | (138, 111) | 48 |
| RGI60-02.05098 | Peyto Glacier | Western Canada and US | 9.69 | (-116.56, 51.65) | (104, 105) | 54 |
| RGI60-01.01104 | Lemon Creek Glacier | Alaska | 9.52 | (-134.35, 58.38) | (75, 125) | 53 |
| RGI60-01.09162 | Wolverine Glacier | Alaska | 16.74 | (-148.90, 60.41) | (96, 122) | 67 |
| RGI60-01.00570 | Gulkana Glacier | Alaska | 17.56 | (-145.42, 63.28) | (132, 103) | 69 |
| RGI60-04.07051 | - | Arctic Canada South | 58.21 | (-80.31, 73.52) | (102, 185) | 117 |
| RGI60-07.00274 | Edvardbreen | Svalbard | 61.18 | (17.57, 77.88) | (132, 133) | 120 |
| RGI60-07.01323 | Biskayerfonna | Svalbard | 12.72 | (12.28, 79.79) | (80, 122) | 60 |
| RGI60-01.17316 | Twaharpies Glacier | Alaska | 54.66 | (-142.08, 61.36) | (195, 109) | 114 |
| RGI60-07.01193 | Skaugumbreen | Svalbard | 8.36 | (14.72, 79.54) | (129, 116) | 50 |
| RGI60-01.22174 | Buckskin Glacier | Alaska | 46.46 | (-150.45, 62.98) | (222, 93) | 105 |
| RGI60-14.07309 | West Ching Kang Glacier | South Asia West | 30.30 | (75.9869, 35.4805) | (118, 135) | 87 |
| RGI60-15.10261 | - | South Asia East | 3.42 | (85.788, 28.404) | (85, 138) | 36 |

**Table C1.** Table of glaciers used for training the UDE. Grid size and Grid res (i.e. resolution) indicate the adaptive constant grid used by OGGM to adapt all gridded data for each glacier.

*Acknowledgements.* We would like to thank Kurt Cuffey for valuable discussions and comments on glacier modelling and physics. Harry
545 Zekollari for all the conversations and help related to large-scale glacier modelling. Per-Olof Persson for the discussions on differential equations and numerical solvers. Giles Hooker for useful feedback regarding the statistical analysis. Chris Rackauckas for all the insights and discussions regarding scientific machine learning in the Julia ecosystem; and the Julia community for the technical support, bug hunting and the interesting discussions in the Julia Slack and Discourse. The Jupyter meets the Earth team (Erik Sundell, Yuvi Panda and many others) for helping with the infrastructure of the Jupyter Hub. J.B. would like to thank the Institut des Géosciences de l'Environnement in
550 Grenoble (France) for hosting him during the whole duration of the project.

J.B. and B.W. were funded by NWO VIDI Grant 016.Vidi.171.063.

We benefited from the Jupyter meets the Earth project supported by the NSF Earth Cube Program under awards 1928406, 1928374.



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
