# Peer review of "Universal Differential Equations for glacier ice flow modelling"

_Geoscientific Model Development, 2023_

## Referee Comment (RC1)

**Review of 'Universal Differential Equations for glacier ice flow modelling' by Bolibar et al.**

Doug Brinkerhoff

July 7, 2023

**1 Summary**

In this paper, the authors describe embedding a neural-network-based parameterization of the viscous rate factor within a shallow ice model and training it on a semi-synthetic dataset with the aid of reverse mode automatic differentiation. I think that this is generally an important topic and that this paper is a useful step in the direction of modelling/ML hybrids. I don't have many objections with respect to the technical content of this work, however I do think it suffers from a few significant misunderstandings of its own methods, of referenced works, and of the broader context into which these results fit. These issues, along with some more minor technical corrections, are outlined below.

**2 Comments**

**L40** It should be noted here that adjoints have been known about and used in glaciology since at least Doug Macayeal's 1993 paper on optimal control methods (MacAyeal, Douglas R. "A tutorial on the use of control methods in ice-sheet modeling." Journal of Glaciology 39.131 (1993): 91-98.). While utilizing neural networks to do things in glaciology is a bit new, the general notion of 'reverse mode AD' is not. (It would also be worth citing Tarasov (2012, "A data-calibrated distribution of deglacial chronologies for the North American ice complex from glaciological modeling." Earth and Planetary Science Letters 315 (2012): 30-40.) as an early example of NN surrogates in glaciology.

**L56 − 64** I find this paragraph to be quite confusing on account of the use of 'scalar parameters'. Many inversion techniques (including the Macayeal paper listed above) are built around finding spatially distributed parameter fields, which are not scalar. Indeed, different forms of regularization in such fields often correspond to a Gaussian process functional prior over spatial (and in some cases temporal) parameter fields. Furthermore, what does 'reduced to the current structure of the mechanical model mean'? As a side note the reference to Brinkerhoff (2016) is questionable: that work uses no gradients but rather very basic MCMC to find distributions over parameter values. However, there are many other good examples of differentiable ice flow models.

**L67** I think it's perhaps a bit of a stretch to say that the neural networks in this work learn 'spatiotemporal variability'. What they are learning is a parameterization of ice softness as a function of surface temperature based on a number of training examples that happen (or are designed) to span a practical domain and range of said function.

**L84** I think it's too early to talk about assuming that $C = 0$ here: the equation remains a "diffusion" equation regardless of this choice and it is not necessary in order to be able to define a $D$.

**L99** 'propriety' → 'property'

**Eq. 4** I suggest using **u** or some other symbolic choice to make clear that velocity is a vector (rather than $V$).

**Sec. 2.2** I think that this section needs to be moved after 2.3, since it is not yet clear what the embedded neural net is supposed to be representing in this case. Furthermore, the function SIASolver is never defined. Does this yield a thickness or a velocity? Since ice velocities are prognostic, why should this function take $V_0$ as an argument, when there cannot be any dependency on this argument? Should this be an $H_0$?

**Sec. 2.2** I really struggle with casting this problem as one that is time dependent and I am not sure I see the value in doing so in this work. The reason for this is that the inversion targets are velocities, which are diagnostic of thickness. Since thickness at some time is being considered as known, then one can simply compute the velocity at this time and use it in computing a loss, updating the parameters of the embedded neural network parameterization of hardness, etc. There is simply no need for a time dependent solver in the procedure as written (although, to be sure there is utility to time-dependent adjoints of glacier models). I think that the consequences of this independence are readily apparent in the authors results in the sense that they find no trouble in recovering their chosen ice hardness parameterization even with mis-specified surface mass balance rates: the reason for this is that the mass balance doesn't really affect the velocity and thus doesn't affect the recovered parameterization. I think that the authors allude to this themselves around L324, but I would like to see some more robust justification for why all of the fancy stuff is necessary here. Finally, as a suggestion for an augmentation that might make all of this a little bit more compelling, is it possible to do all of this with thickness as the predicted value rather than velocity?

**L140** I think that the phrase 'spatial' should be dropped here. This is trying to recover $A(T_s)$. One could imagine a circumstance in which $T_s$ is a function of space internally to each training example, but it does not seem to be the case here (based on Fig. 3, although I could misunderstand this).

**L144** I think that this part is confusing: the authors write that their function for $A$ ignores ice temperature, but isn't that what the surface temperature is supposed to proxy? This section is generally a bit unclear and could benefit from improved notation. For example, I suggest using the symbol $T_s$ for surface temperature, to make it clear that this is what is being fed into the parameterization.

**L162** Why 'timestamps' and not just 'times'?

**Sec 3.1** Typically model runs require some type of spinup because the physics and various data products are not all self-consistent, leading to unrealistically large transient behavior as all of these things equilibrate. Was that the case here? Does it influence the results?

**L176** I know that the CFL is mentioned in Appendix A, but it would be worth summarizing the methods used to ensure time-stepping stability conditions are satisfied here.

**177–180** Why is noise applied to $A$ rather than to the observed velocity? It's not $A$ that is being observed, and thus it makes little sense to simulate noise in $A$.

**L181** Most readers will not be able to infer what (1,3,10,3,1) means when describing neural networks. This either needs to be expanded here, or a discussion of neural network architecture choice should be added as an appendix.

**L182** Once again, I think that some clarification of what is actually happening here is in order: the authors are using a very common formulation of a numerical ice flow model in a standard way. The novelty here is the replacement of a static rate factor $A$ with a function $A_\omega(T_s)$ that is parameterized via a function that happens to be quite flexible (a neural network). I don't think that this is semantically equivalent to saying the problem is highly constrained by the PDE nature of the ice flow model.

**L186** Why is the final sigmoid necessary? What happens without it? I can understand the desire to impose positivity (which could perhaps more easily be effected by log-transforming $A$, as is commonly done), but why should there be an upper bound?

**L195–198** I don't understand this section. Why does adding a source term in the ODE produce instability? What is the $H$ matrix? Are you saying that you're adding the mass balance after the integration of the flow equations?

**L199–205** How long are these runs for? Is there sufficient geometric change over this time period to warrant such a detailed approach (and are the changes large enough to significantly affect $A$)? Shouldn't $A$ respond over fairly long time scales, since it's ultimately ice temperature rather than surface temperature that controls flow?

**Sec 3.4** I think that this section is too deep with respect to AD to be of use to glaciologists reading this, while being too shallow to be of use to AD practitioners. I suggest either delving deeply into a software-engineering type study of the influence of different AD approaches or to cut this section and simply describe the approach that was actually used. This goes for Appendix B as well, which is mostly a textbook definition of finite differences and the adjoint method.

**L254–255** Referencing the potential broader impacts of cloud computing is fine, but a statement like this needs to be backed up with some evidence: it is not immediately obvious that the cloud improves scientific equity.

**Fig. 3** This figure begs the question: why use a neural network for parameterizing $A(T)$. It is clear that a quadratic or exponential would have worked just as well. Indeed, if the authors had predicted the log of $A$, then it is likely that a linear model would have fit these data just as well as the NN. I appreciate the potential for generalization of the NN approach, but an ablation study with simpler models might be helpful here.

**L4.1** Again, because velocity is diagnostic, this is not a surprising outcome.

**L316** There is a substantial literature on the joint inference of traction and rheological parameters (typically in a Bayesian framework which allows for a quantification of induced covariance between parameters). In summary, this inversion is not necessarily ill-posed 'by nature' because there is scale separation between different processes.

**L324** This description of the seasonal cycle of glacier velocities is perhaps oversimplified: many glaciers exhibit minimal velocities at the end of summer and speed up during the winter. Maybe reword to express a bit more nuance?

**L329** I don't understand the use of 'initial conditions' here. If we're not doing time evolution, then the conditions aren't really 'initial', they're just the geometry.

**L335** It would be nice to see some references that illustrate the so-called 'equifinality problem'.

**Sec. 5** Throughout this work, the uncertainty in inferred parameters is not addressed. I think that there should at least be a discussion of the potential implications of such and avenues for providing a more rigorous uncertainty quantification.

**Sec. 5.2.1** This editorial on AD approaches is not relevant to the current work.

**L374** The authors seem to have a misunderstanding of PINNs, which involve positing a neural network as the PDE solution and then using a point collocation method to adjust the solution so as to minimize the solution residual in some norm. As such, a PINN is *not* a surrogate, but rather is a numerical method for solving PDEs in the same vein as FEM or FD and does not require 'training data' in the way that it is usually understood in the ML literature. In contrast, the work of Jouvet (which is indeed

a surrogate, but which does not employ PINNs in the described way) trains a CNN to operate as the approximate solution operator to the ice flow equations from many examples of solutions generated by a 'normal' ice flow model (or perhaps by using a PINN).

**Sec. 5.2.3** I do not understand this section.

**Eq. A9,A10** I don't understand the how these conditions related to step size choices. Is there another reference that might illustrate this point more clearly?

---

## Author Comment (AC1)

**Review response for *Universal Differential Equations for glacier ice flow modelling**

**The Authors**

*September 20, 2023*

**Dear Reviewer Douglas Brinkerhoff,**

We thank you for your constructive comments and suggestions. Here we have elaborated on some of your points. Original *reviewer responses are in italic*, while the authors responses can be found in blue.

*In this paper, the authors describe embedding a neural-network-based parameterization of the viscous rate factor within a shallow ice model and training it on a semi-synthetic dataset with the aid of reverse mode automatic differentiation. I think that this is generally an important topic and that this paper is a useful step in the direction of modelling/ML hybrids. I don't have many objections with respect to the technical content of this work, however I do think it suffers from a few significant misunderstandings of its own methods, of referenced works, and of the broader context into which these results fit. These issues, along with some more minor technical corrections, are outlined below.*

**1. COMMENTS**

L40 *It should be noted here that adjoints have been known about and used in glaciology since at least Doug Macayeal's 1993 paper on optimal control methods (MacAyeal, Douglas R. "A tutorial on the use of control methods in ice-sheet modeling." Journal of Glaciology 39.131 (1993): 91-98.). While utilizing neural networks to do things in glaciology is a bit new, the general notion of 'reverse mode AD' is not. (It would also be worth citing Tarasov (2012, "A data-calibrated distribution of deglacial chronologies for the North American ice complex from glaciological modeling." Earth and Planetary Science Letters 315 (2012): 30-40.) as an early example of NN surrogates in glaciology.*

We completely agree with this point. We believe we have attempted to address this point in L51, where we try to make the difference between what efforts have been made so far in terms of differentiable programming in glaciology with respect to this study. We have added the following sentence in that paragraph to add this context: *"Such gradients can be found by either computing the associated adjoint or by using AD."*. We are also grateful for the references.

L56–64 *I find this paragraph to be quite confusing on account of the use of 'scalar parameters'. Many inversion techniques (including the Macayeal paper listed above) are built around finding spatially distributed parameter fields, which are not scalar. Indeed, different forms of regularization in such fields often correspond to a Gaussian process functional prior over spatial (and in some cases temporal) parameter fields. Furthermore, what does 'reduced to the current structure of the mechanical model mean'? As a side note the reference to Brinkerhoff (2016) is questionable: that work uses no gradients but rather very basic MCMC to find distributions over parameter values. However, there are many other good examples of differentiable ice flow models.*

We agree that this concept is not easy to communicate, and it might require some improvements in the way we are writing it. What we mean by this, is the fact that in what we call "scalar inversions" the structure of the model (i.e. the equation itself) is not modified. In this type of inversions, parameters already present in the equation are fitted. However, for the case of functional inversions, we are actually inverting a function, which can then be translated into a mathematical expression, thus expanding the already existing equation. This implies that such inversions "build" on top of already existing models, expanding them by adding new parts to them. In order to convey this point, we have restructured this paragraph in the following way:

"*Nonetheless, all efforts so far have been applied to the inversion of scalar parameters and sometimes their distributions, i.e. parameters that are stationary for a single inversion given a dataset. This means that the potential of learning the underlying physical processes is reduced to the current structure of the mechanistic model. No changes are made to the equations themselves, with the main role of the inversions being the fitting of one or more parameters already present in the equations. To advance beyond scalar parameter inversions, more complex inversions are required, shifting towards functional inversions. Functional inversions enable the capture of relationships between a parameter of interest and other proxy variables, resulting in a function that can serve as a law or parametrization. These learnt functions can then be added in the currently existing equation, thus expanding the underlying model with new knowledge.*"

L67 *I think it's perhaps a bit of a stretch to say that the neural networks in this work learn 'spatiotemporal variability'. What they are learning is a parameterization of ice softness as a function of surface temperature based on a number of training examples that happen (or are designed) to span a practical domain and range of said function.*

Agreed, but we believe it is a good way to convey the fact that any empirical law will ultimately depend on some proxies, which in the context of the geosciences, vary in the temporal and spatial dimensions. Of course the dependency is not on space itself, but in this case the climate does depend on space (and also time). So actually, what the

NN is learning is the **spatial** variability (i.e. between glaciers) of $A$. It is also capturing a temporal variability, but since the climate is computed with a 30 year rolling mean, the changes are almost imperceptible.

L84 *I think it's too early to talk about assuming that $C = 0$ here: the equation remains a "diffusion" equation regardless of this choice and it is not necessary in order to be able to define a D.*

We agree that the equation is a diffusivity equation independently of the assumption $C = 0$. We have now introduced the title of *diffusivity equation* before the assumption $C = 0$, but we believe this simplification is useful to introduce the more compact form of the SIA equation we use for this application.

L99 *'propriety' $\rightarrow$ 'property'*

Noted.

Eq. 4 *I suggest using u or some other symbolic choice to make clear that velocity is a vector (rather than V).*

Thanks for the suggestion. We have changed the notation to $u$.

Sec. 2.2 *I think that this section needs to be moved after 2.3, since it is not yet clear what the embedded neural net is supposed to be representing in this case. Furthermore, the function SIASolver is never defined. Does this yield a thickness or a velocity? Since ice velocities are prognostic, why should this function take V0 as an argument, when there cannot be any dependency on this argument? Should this be an H0?*

We chose to present the concept of UDEs before presenting the actual functional inversion we use in this study in order to first understand the global concept before introduce an example of it. The way we present UDEs, and $D_\theta$ is independent from what that function is actually parametrising. First we introduce the big idea, and then in section 2.3 we introduce an application of this idea.

We have added a definition for SIASolver. SIASolver yields a thickness, but then by performing a simple transformation we can obtain the surface velocities, as explained in Equation 4. We have clarified all this in the following manner: *"For a single glacier, if we observed two different ice surface velocities $u_0$ and $u_1$ at times $t_0$ and $t_1$, respectively, then we want to find $\theta$ that minimizes the discrepancy between $u_1$ and $SIASolver(H_0, t_0, t_1, D_\theta)$, defined as the forward numerical solution of the SIA equation yielding a surface ice velocity field following Equation 4"*

Regarding $u_0$, indeed, this should be $H_0$, since this is what is used by SIASolver as initial conditions. We have updated this accordingly.

Sec. 2.2 *I really struggle with casting this problem as one that is time dependent and I am not sure I see the value in doing so in this work. The reason for this is that the inversion targets are velocities, which are diagnostic of thickness. Since thickness at some time is being considered as known, then one can simply compute the velocity at this time and use it in computing a loss, updating the parameters of the embedded neural network parameterization of hardness, etc. There is simply no need for a time dependent solver in the procedure as written (although, to be sure there is utility to time-dependent adjoints of glacier models). I think that the consequences of this independence are readily apparent in the authors results in the sense that they find no trouble in recovering their chosen ice hardness parameterization even with mis-specified surface mass balance rates: the reason for this is that the mass balance doesn't really affect the velocity and thus doesn't affect the recovered parameterization. I think that the authors allude to this themselves around L324, but I would like to see some more robust justification for why all of the fancy stuff is necessary here. Finally, as a suggestion for an augmentation that might make all of this a little bit more compelling, is it possible to do all of this with thickness as the predicted value rather than velocity?*

This is a very good point. We tried to argue about these aspects in section 4.1 and in the discussion (L326-330). Regarding the use of V against H. Indeed, we have tried to use H as the target for the inversions, but this seems to work only when no surface mass balance is present. The mass balance alters the ice thickness, thus introducing too much noise for the model to correctly invert $A$. On the other hand, V is not sensitive to the surface mass balance, as we have shown in this study. Therefore, we believe V is a much better target data for inversions regarding ice rheology in the presence of a surface mass balance signal.

As we explained in L326-330, we do agree that this approach might seem quite overkill for the problem at hand. But again, like the use of a NN, this is to prove that this more complex setup can work with this more complex nonlinear PDEs. Nonetheless, some preliminary tests trying to solve this problem with a time-independent method (not shown in this study) showed that that sort of inversion is highly sensitive to noise. Small changes in the surface conditions (i.e. due to mass balance), made it quite hard for the inversion to easily converge. Since we believe the more complex approach with a time-dependent solver has more potential, both in terms of data assimilation in the presence of noise and for transient simulations, we decided to focus on this.

In order to clarify this, we have added the following sentence in the sub-section "Robustness to noise in observations": *"Various tests using H instead of u as the target*

*data showed that A cannot be successfully inverted in the presence of a surface mass balance signal."*

L140 *I think that the phrase 'spatial' should be dropped here. This is trying to recover A(Ts). One could imagine a circumstance in which Ts is a function of space internally to each training example, but it does not seem to be the case here (based on Fig. 3, although I could misunderstand this)*

As explained above, we do believe that predicting the relationship with respect to climate accounts for spatial dependencies. Here, the climate is calculated at each glacier's centroid. This means that the climate signal changes through space (i.e. among glaciers). When referring to spatial dependencies, we are not referring to spatial changes **within** a glacier, but **among** glaciers (i.e. in their coordinates).

We have added this nuance in the text: *"The objective of $A_\theta(T)$ will be to learn the spatial variability (i.e. among glaciers) of A with respect to T for multiple glaciers in different climates."*

L144 *I think that this part is confusing: the authors write that their function for A ignores ice temperature, but isn't that what the surface temperature is supposed to proxy? This section is generally a bit unclear and could benefit from improved notation. For example, I suggest using the symbol Ts for surface temperature, to make it clear that this is what is being fed into the parameterization.*

Indeed, it ignores ice temperature, since it's not an input feature of the NN. We use surface air temperature as a proxy of ice temperature. As suggested, we have updated all instances of $T$ to $T_s$ Moreover, we have rephrased this sentence to make this point clearer: *"This relationship is based on the hypothesis that $T_s$ is a proxy of ice temperature, and therefore of A. However, it ignores many other important physical drivers influencing the value of A, such as a direct relationship with the temperature of ice, the type of fabric and the water content."*

L162 *Why 'timestamps' and not just 'times'?*

We agree that the word times here is mode adequate.

Sec 3.1 *Typically model runs require some type of spinup because the physics and various data products are not all self-consistent, leading to unrealistically large transient behavior as all of these things equilibrate. Was that the case here? Does it influence the results?*

This is true. Nonetheless, for this case study we did not experience any initial shock in the simulations that we had to deal with. Both the used initial ice thickness and the transient simulation look quite smooth.

L176 *I know that the CFL is mentioned in Appendix A, but it would be worth summarizing the methods used to ensure time-stepping stability conditions are satisfied here.*

Thank you for the suggestion. We have added the following sentence explaining this in the manuscript based on the current implementation of the numerical solver: *"The method implements an adaptive temporal step size close to the maximum value satisfying the CFL conditions for numerical stability at the same time that controls numerical error and computational storage"*

177–180 *Why is noise applied to A rather than to the observed velocity? It's not A that is being observed, and thus it makes little sense to simulate noise in A.*

This is a good point. We could have directly added the noise in the surface velocity. However, in practical terms this accounts to almost the same. Perturbating $A$ results in surface velocity perturbations, so the result is equivalent. Moreover, since what we want to learn is $A$, this also gives us insight on how close we can get to a noisy empirical law. At the end of the day, empirical laws are just a fitted function minimising the error. There is indeed a transformation from $A$ to $u$, but we find it interesting to directly treat the $A$ space, since it related more with the "fundamental" function we are trying to learn.

L181 *Most readers will not be able to infer what (1,3,10,3,1) means when describing neural networks. This either needs to be expanded here, or a discussion of neural network architecture choice should be added as an appendix.*

We agree with this point. We had added a more detailed explanation regarding the architecture without relying in the extra notation.

L182 *Once again, I think that some clarification of what is actually happening here is in order: the authors are using a very common formulation of a numerical ice flow model in a standard way. The novelty here is the replacement of a static rate factor A with a function $A_\omega(Ts)$ that is parameterized via a function that happens to be quite flexible (a neural network). I don't think that this is semantically equivalent to saying the problem is highly constrained by the PDE nature of the ice flow model.*

I think our original point was not correctly communicated. What we intended to explain here is that this approach is simpler than a classic data-driven formulation of the problem, where the neural network would have to learn the whole dynamics of the problem directly from data, from a purely machine learning perspective. The fact that we use a numerical solver to solve a differential equation, and we leave just a small subset of the dynamics to the neural network, results in a much simpler learning problem, since we are imposing the optimization problem with the structure of the SIA. We have updated

the text by explicitly mentioning this:

*"Since the optimization problem is much more constrained by the structure of the solutions of the PDE compared to a pure data-driven approach, a very small neural network is enough to learn the dynamics related to the subpart of the equation it is representing (i.e. A)."*

L186 *Why is the final sigmoid necessary? What happens without it? I can understand the desire to impose positivity (which could perhaps more easily be effected by log-transforming A, as is commonly done), but why should there be an upper bound?*

This sigmoid output function with capped values is necessary for the NN to avoid producing $A$ values that are too large. If an upper bound is not used, the NN can output large values (i.e. much larger than $10^{-15}$), which when "injected" into the SIA produce numerical instabilities (because the minimum and maximum timestep in the solver may be also constrained) or makes the explicit solver to take smaller timesteps to ensure suitability, which then leads to more computationally expensive computation of the gradients. Moreover, this is part of the philosophy of introducing any possible physical constrain into the model. If we know *a priori* that the values of $A$ have a given range, it makes sense to include this into the model.

In order to clarify this, we have added the following sentence: *Constraining the output values of the neural network is necessary in order to avoid numerical instabilities in the solver or very small stepsizes in the forward model than will lead to expensive computations of the gradient.*

L195–198 *I don't understand this section. Why does adding a source term in the ODE produce instability? What is the H matrix? Are you saying that you're adding the mass balance after the integration of the flow equations?*

The way the solvers in the DifferentialEquations.jl library are coded, imply that one can seamlessly change the solver used to solve a given differential equation. This means that the time stepping mechanisms are hidden under the hood to the user. For the solvers to work correctly, it is important not to add any substantial source during the computation of a time step, since it can interfere with the internals of each solver. We observed empirically the effect of numerical instabilities when adding the mass balance contribution at every step size in the solver. In order to add the contribution of mass balance to the ice thickness we decided instead to add it in discrete steps every month, which makes sense both computationally and numerically. For that, we used DiscreteCallback, which is especially designed to handle events during the solving of a differential equation. They get called when a given condition is met. E.g. for the mass balance, once a month of time has elapsed.

We have updated this sentence in order to clarify this and the reference to the "H matrix": *"In order to add the surface mass balance term $\dot{b}$ in the SIA Equation we used a `DiscreteCallback` from `DifferentialEquations.jl`. This enabled the modification of the glacier ice thickness $H$ with any desired time intervals and without producing numerical instabilities."*

L199–205 *How long are these runs for? Is there sufficient geometric change over this time period to warrant such a detailed approach (and are the changes large enough to significantly affect A)? Shouldn't A re- spond over fairly long time scales, since it's ultimately ice temperature rather than surface temperature that controls flow?*

In this study, we mostly ran simulations for a period of 5 to 10 years. These periods are much shorter than the whole W5E5 or ERA5 period available in the raw file given by OGGM. Therefore, in terms of memory usage, it is much better to preprocess once the file, dropping unnecessary variables and cropping the needed time period, than doing so each time step (i.e. monthly) in the solver. This is not done only to drive changes in $A$. The main reason behind this is the computation of the surface mass balance, which needs a detailed evolution of the climate on the glacier. That is why the name of the sub-section is called "Surface mass balance".

Sec 3.4 *I think that this section is too deep with respect to AD to be of use to glaciologists reading this, while being too shallow to be of use to AD practitioners. I suggest either delving deeply into a software- engineering type study of the influence of different AD approaches or to cut this section and simply describe the approach that was actually used. This goes for Appendix B as well, which is mostly a textbook definition of finite differences and the adjoint method.*

We agree with the reviewer about the fact that here we are dealing with different audiences. Rather than targeting the glaciologists or general AD practitioners, here we focus on the set of practitioners working in the Scientific Machine learning ecosystem in Julia. There is a large amount of people currently attempting to use these tools for complex problems in geophysics. Focusing on these aspects is useful for other researchers working in different applications, as we have experienced in conversations with people in the community. As mentioned in the abstract, we believe this is one of the interesting bits of information that a lot of people who will read the paper will be looking for.

L254–255 *Referencing the potential broader impacts of cloud computing is fine, but a statement like this needs to be backed up with some evidence: it is not immediately obvious that the cloud improves scientific equity.*

We thank the reviewer for rising this point. Although we the authors have experienced the benefits of cloud computing in terms of scientific and pedagogical inclusivity, unfortunately there is no formal study assessing the validity of this statement. We hope to see work in this line in the future. We have then decided to remove this sentence from the manuscript.

Fig. 3 *This figure begs the question: why use a neural network for parameterizing A(T). It is clear that a quadratic or exponential would have worked just as well. Indeed, if the authors had predicted the log of A, then it is likely that a linear model would have fit these data just as well as the NN. I appreciate the potential for generalization of the NN approach, but an ablation study with simpler models might be helpful here.*

This is indeed true. As you mentioned, the main reason behind to choice of a NN is its universality in terms of function representation plus their capacity of being easily differentiable with respect to their parameters. We purposely chose a simple case in order to evaluate the performance in a controlled environment. For any other real case scenario, any family of regression function with good expressively will serve well, but NN are particularly easy to train inside the SciML ecosystem in Julia. It is important to emphasize that NNs are easy to differentiate, compared to many other methods (e.g. including tree-based methods). These aspects are largely commented in Rackauckas et al. (2020).

L4.1 *Again, because velocity is diagnostic, this is not a surprising outcome.*

Agreed

L316 *There is a substantial literature on the joint inference of traction and rheological parameters (typically in a Bayesian framework which allows for a quantification of induced covariance between parameters). In summary, this inversion is not necessarily ill-posed 'by nature' because there is scale separation between different processes.*

Thanks for this comment. We have adjusted the text in order to nuance a bit more this message in the following way: *"Nonetheless, despite a scale difference between these two processes, this can be an ill-posed problem, since the only available ice velocity observations are from the surface, encompassing both creep and basal sliding."*. Indeed, this is an interesting point to explore in future work, also to evaluate different methods that can do inference on these two different contributions to the diffusivity.

L324 *This description of the seasonal cycle of glacier velocities is perhaps oversimplified: many glaciers exhibit minimal velocities at the end of summer and speed up during the winter. Maybe reword to express a bit more nuance?*

Indeed, this is simplified. We don't want to get into details about these processes here. Nonetheless, for midlatitude temperate glaciers, this is the main trend. In fact, here we are not talking about velocities, we are just mentioning the main contributions of creep during winter, and the onset of sliding during summer (independently of the effects on surface velocity).

L329 *I don't understand the use of 'initial conditions' here. If we're not doing time evolution, then the conditions aren't really 'initial', they're just the geometry.*

Good point. We have updated this term to "geometry".

L335 *It would be nice to see some references that illustrate the so-called 'equifinality problem'.*

We have added two references covering this equifinality problem.

Sec. 5 *Throughout this work, the uncertainty in inferred parameters is not addressed. I think that there should at least be a discussion of the potential implications of such and avenues for providing a more rigorous uncertainty quantification.*

Indeed, uncertainty quantification is not addressed in this work. Although we recognize this is an important piece in geophysical models and that informing about the confidence we have in our estimates is useful, we decided to leave this point of discussion for future work. Notice than in this setup uncertainty quantification needs to be carried on the output of the NN, and it's not uncertainty on the parameters of the same. Therefore, this implies inferring the uncertainty of the learnt function.

Sec. 5.2.1 *This editorial on AD approaches is not relevant to the current work.*

This paper has two distinct target audiences: (1) Computational glaciologists interested in advanced techniques to make inversions to improve understanding of glacier physical processes; (2) Computer scientists and statisticians interested in Universal Differential Equations and the Julia ecosystem. This subsection is directly targeted to the second audience. It might seem out of topic for the glaciology community, but this information is highly relevant to Julia practitioners who are interested in using these methods for their own research.

L374 *The authors seem to have a misunderstanding of PINNs, which involve positing a neural network as the PDE solution and then using a point collocation method to adjust the solution so as to minimize the solution residual in some norm. As such, a PINN is not a surrogate, but rather is a numerical method for solving PDEs in the same vein as FEM or FD and does not require 'training data' in the way that it is usually understood in the ML literature. In contrast, the work of Jouvet (which is indeed a surrogate, but which does not employ PINNs in the described way) trains a CNN to operate as the*

*approximate solution operator to the ice flow equations from many examples of solutions generated by a 'normal' ice flow model (or perhaps by using a PINN).*

We thank the reviewer for this comment. We agree that the two concepts of using NN as numerical solver and emulators are explained in a rather confusing way in this section. The connection here with PINNs is confusing and we rather wanted to refer to the family of emulators that can be used to emulate a numerical solver for different initial condition and parameter choices. We have removed the reference to PINNs and keep the one to emulators.

Sec. 5.2.3 *I do not understand this section.*

We agree that this section it was difficult to read as it was presented. We have re-written this section to make our point more clear and understandable.

Eq. A9,A10 *I don't understand the how these conditions related to step size choices. Is there another reference that might illustrate this point more clearly?*

This condition is not related to the choice of the stepsize. Instead, this condition is used to force the condition that the ice thickness cannot take negative values. We apologize for not including a further reference for this point. We have included the pertinent reference to the manuscript: *Imhof, M. A.: Combined climate-ice flow modelling of the Alpine ice field during the Last Glacial Maximum, VAW-Mitteilungen, 260, 2021*

---

## Author Comment (AC2)

**Review response for Universal Differential Equations for glacier ice flow modelling**

**The Authors**

*September 20, 2023*

**Dear Reviewer ,**

We thank you for your useful comments and suggestions. Here we have elaborated on some of your points. Original *reviewer response is in italic*, while the authors response can be found in blue.

*The manuscript presents an approach that combines inverse methods ("data assimilation") with machine learning-based parameter estimation using the concept of Universal Differential Equations (UDEs) or Neural Ordinary Differential Equations (N-ODEs). The approach has the potential to enable a wider and more rigorous use of diverse observations to constrain or calibrate physical models. For it to work, the notion of differentiable programming becomes key, i.e., the PDE representing the physical model and the UDE embedded within the PDE need to be "differentiable" in the sense that an adjoint model needs to be generated. This adjoint model serves to compute the gradient of the "cost" or "loss" function with respect to uncertain/unkown parameters. The gradient, in turn, is an essential ingredient for gradient-based optimization. A versatile approach to generate the adjoint of the PDE/UDE system is by means of automatic differentiation. The approach is demonstrated by calibrating the glacier flow model ODINN.jl on a range of glacier observations from the Randolph Glacier Inventory. The example is simple, meant as a prototype demonstration of the method. The PDE consists of the SIA model for glacier flow. The parameterization within the SIA to be improved upon is Glen's creep parameter A and its functional representation on the climate temperature normal T. Parameter(ization) A is replaced by a neutal network (NN), and the task is to estimate the NN parameters θ. The manuscript concludes with a discussion of challenges and opportunities.*

*The manuscript presents an exciting approach of combining physics-based inverse modeling with machine learning-based approaches to improve the way that observational data may be used to rigorously constrain or calibrate models. The seamless integration of PDEs and UDEs for model calibration or parameter learning represents in my view a promising applications of machine learning for surrogate modeling in combination with physics-based models. I recommend publication after minor revisions. My comments are largely intended to improve the manuscript's clarity.*

**Main (but minor) comments:**

- *line 136-137 and eqn. (3) "Instead of considering that the diffusivity $D(\theta)$ is the output of a universal approximator, we are going to replace the creep parameter $A$ in Equation (3)..." I agree with this approach, but perhaps a sentence to motivate this choice might be warranted.*

  We explained this in the last paragraph of the section with the following sentence: "Nonetheless, this simple example serves to illustrate the modelling framework based on UDEs for glacier ice flow modelling, while acting as a platform to present both the technical challenges and adaptations performed in the process, and the future perspectives for applications at larger scales with additional data.".

- *Also, eqn. (7) seems to be the crux in connecting the PDE with the UDE. It might be worth, then, to be very explicit that $A(\theta)$ becomes the surrogate model embedded within the PDE. To connect eqn. (7) with the general eqn. (5), perhaps relate $A(\theta)$ with the universal approximator $U(\theta)$ defined in eqn. (5).*

  Thank you for the suggestion, we have added an observation with this point.

- *line 141: "...by prescribing an artificial law for which a reference dataset is generated..." This is perhaps somewhat imprecise. As stated, it gives the impression that a reference dataset is generated for $A$, when in fact, the actual data set to be used (and generated via $A$) are the velocities obtained as solution and used in the loss function.*

  We agree this is rather confusing. We have rephared this sentence and added it in the following format: "Given this artificial law for $A(T_s)$, a reference dataset of velocities and ice thicknesses is generated by solving the SIA equation (2)".

- *lines 142/142: "...we have used the relationship between ice temperature and $A$ from Cuffey and Paterson (2010), and replaced ice temperatures with a relationship between $A$ and $T$" This is a crucial methodological step in this work to generate the reference data. This step should be described more clearly. I think what is done here is to provide a synthetic solution for a known functional form $A(T)$ described in Cuffey and Patterson (2010), which then will be attempted to be inferred. I feel this relationship should be reproduced here to make this paper more accessible and self-contained.*

  We thank the reviewer for the suggestion of making the synthetic relationship between $A$ and $T$ more explicit in the manuscript. However, we believe the actual relationship between these variables is not really important to illustrate the behaviour of our model and could mislead the reader on the specifies of this relationship. Furthermore, it is important here to remark that we are using proxies for the ice temperature and not the

ice temperature directly, making the decision of the fake law even more arbitrary for actual scientific purposes.

- *Figure 2: Perhaps two such schematics should be drawn, one using the functional form of A as taken from Cuffey and Patterson (2010) to generate the reference data (with added noise to A), and the other one as is(?)*

  As we discussed in the following point, we believe it is much simpler to keep the current figure, which easily encapsulates both concepts, and gives a more general overview of the modelling philosophy.

- *The box for the "Physical law": wouldn't it make sense to state A as a function of the NN, instead of D, because the rest of the functional form of D remains unchanged?*

  Even if the part of the diffusivity $D$ that is being modelleled using a neural network is just $A$, we believe this formulation is more general and communicates better the idea of the model. This scheme is more general and can accommodate more variations of the presented parameterization.

- *Also, why is there an arrow from the PDE solver to the Mass balance box?*

  The reason for this is that mass balance models depends of the ice surface elevation: higher altitudes are associated to colder climates with more accumulation. Then, the different values of the diffusivity $D_\theta$ will lead to different solutions of ice thickness $H$ that then will have to be fed again in the mass balance model to recalculate the accumulation/ablation rate. This step is essential to correctly take into account the topographical mass balance feedback explained in Zekollari et al. (2020) and Bolibar et al. (2022).

- *line 223 and again 523: "checkpointing": Accurate checkointing is not actually an interpolation between stored checkpoint. AD-related checkpointing methods are designed to balance storing vs recomputation with the aim to recover the required state exactly. Griewank and Walther (2008), chapter 12 cover this. In the context of the latest implementation in Julia, see Schanen et al. (2023).*

  We thank you for the references and observations for this section, we have added them to the manuscript. We have changed the phrasing of these two sentences based on this observation.

- *Equation (8): A typo denominator, should be $\nabla_\theta A$ ("A" missing)*

  Noted.

- *line 350: "Julia source code is automatically differentiable, ...": This is certainly too strong of a statement. While sometimes claimed by members of the core Julia community, it is currently an ambition but which clearly is not (yet) realized in its generality. To move toward this goal requires further developments in general-purpose automatic differentiation. Arguably, the latest developments of Enzyme.jl (Moses et al. 2022) are a promising step in this direction.*

  We completely agree with this point and we agree that the sentence is rather general and not accurate. We have removed this sentence from the manuscript but leaving the discussion on the different AD packages.

**Other minor issues:**

- *line 59-60 and again line 67: The statement "Nonetheless, all efforts so far have been applied to the inversion of scalar parameters, i.e. parameters that are stationary for a single inversion given a dataset." is not quite accurate (or I misunderstood it, in which case it may need clarification). The study by Goldberg et al. (2015; see references) inverts for time-varying and spatially distributed open boundary conditions. You may mean something else though with this sentence(?)*

  This is a good point, also raised by Reviewer 1. Following these two comments, we have rephrased the whole paragraph in the following way:

  *"These inverse modelling frameworks enable the minimization of a loss function by finding the optimal values of parameters via their gradients. Such gradients can be found by either manually coding the adjoint or by using AD. Nonetheless, all efforts so far have been applied to the inversion of scalar parameters and sometimes their distributions, i.e. parameters that are stationary for a single inversion given a dataset. This means that the potential of learning the underlying physical processes is reduced to the current structure of the mechanistic model. No changes are made to the equations themselves, with the main role of the inversions being the fitting of one or more parameters already present in the equations. To advance beyond scalar parameter inversions, more complex inversions are required, shifting towards functional inversions. Functional inversions enable the capture of relationships between a parameter of interest and other proxy variables, resulting in a function that can serve as a law or parametrization. These learnt functions can then be added in the currently existing equation, thus expanding the underlying model with new knowledge."*

- *line 118: The use of the term "observed" may be misleading in the present context. While true in general, the "observed" data used in study are actually simulated reference data (from a known functional relationship). It may be worth clarifying this. Also, to be notationally clear, these "observed"/simulated reference velocities are the elements*

$V_1^k$ in eqn. (6), right?

We have changed "observed" to "target" to be more precise about this point. Regarding the second question: that is correct, the velocities described as observed/target are the $V_1^k$ (denoted as $u_1^k$ in the new version of the manuscript) in Equation 6.

- *line 128: In the case where the $\omega_k$ are the diagonal elements of a covariance matrix, this would essentially amount to a simple weighted least-squares optimization problem (with weights equal to the the inverse variances).*

Indeed, the weighted least square problem can be interpreted as a maximum likelihood model where we include the information about the covariance matrix. However, notice that here the optimization is still carried with respect to the weights of the neural network. However, here the weights $\omega_k = \|u_0^k\|_F$ are not playing the role of variances. Instead, they control the differences in observed velocities due to changes in $A$, so the ideal desired weights should account for variances on $A$ instead of the parameter models. This is an interesting point to explore in future work.

- *line 179: "... output of the prescribed law ...": Just to clarify: the noise is added to the output of the prescribed law $A(T)$, i.e., one generates $A(t) + \epsilon$, with $\epsilon$ being the added noise?*

This is correct.

- *Next sentence: "This setup is used to compute the reference solutions, ...": It might again be worth clarifying that these are the reference solutions on the velocities $V_1^k$ used in eqn. (6).*

We have changed this sentence to "reference synthetic solutions" to emphasize this point. We have also added the reference to the $u_1^k$ in Equation 6 to make the link more clear.

- *Section 3.3 It might be useful to replace everywhere "mass balance" (and "MB") by "surface mass balance" ("SMB").*

Thank you for the suggestion. We have added "surface" in the manuscript when referring to the surface mass balance.

- *line 197: "H matrix": not defined what this refers to.*

The ice thickness $H$ has been introduced in section 2.1 in order to write the SIA equation. We have added "glacier ice thickness $H$" to this sentence for clarification.

- Section 3.4, 3.5: I suggest merging these two sections under the title "Sensitivity methods, differentiation, and optimization" and convert current section 3.5 to 3.4.3.

  Thanks for the suggestion. We believe it might be clearer to keep these separated. First, to avoid too much depth in the subsections, and second because the optimization methods are independent from the sensitivity methods.

- *line 255: "unrepresented" -¿ "under-represented"*

  Noted.

- *line 257:"2i2c": please add a reference*

  We have added a reference for JupyterHub in this section. Unfortunately, there is no direct way or acknowledging 2i2c, so we removed this mention in the manuscript and keeping the reference to 2i2c in the acknowledgement section.

- *line 263: "with respect TO a predictor"*

  Noted.

- *line 270/271: "until it finds an optimal non-overfitted solution". Could you briefly elaborate how you determined that there is no over-fitting.*

  This is a very empirical observation based on the converge plot shown in Figure 3. Here we can observed how the loss function gets smaller but at the same time that the obtained law for $A_\theta(T)$ is smooth. Over-fitting the solution in this context will mean that the NN will over-fit the values of $A(T) + \epsilon$, that is, the values of $A$ with the added noise. The fact that the model does not try to fit the exact values of $A$ is a good indication that the learned curved is robust to noise and it is estimating the actual value of $A(T)$.

- *line 299-301: How should one interpret this? Shouldn't the functional form $A(T)$ be different in both cases then? I.e., the error in the SMB, should be compensated by the NN model of $A(T)$?*

  This is what we mentioned in the discussion. This is mainly due to the robustness of ice surface velocities to changes in surface mass balance. If one used $H$, the ice thickness, then the signal would change too much for the network to be able to recover the function. This robustness is linked to the observations, not so much to the method (i.e. the NN) itself.

- *line 306-308: "This weak dependence...": This is certainly a valid perspective. However, if one generalizes the problem to also use altimetric data to constrain ice height, then*

*this no longer applies and weak dependence on SMB may become problematic(?)*

Indeed, this is a good point. As we mentioned in the point above, this is good news for rheology/basal conditions inversions IF ice velociy data is available. Altimetry data could also be used in conjunction (we haven't investigated this, though), but it would then be affected by the SMB signal.

- *Subsubsection 5.2.1: Automatic differentiation is independent of SciML. I would give it its own subsection.*

  The main idea behind this division is to divide the discussion in the two main topics of our research: (1) on the one hand, we have glaciology, and the domain-specific science related to the physical laws and the solving of the differential equations. (2) On the other hand we have the Scientific machine learning topics, which in our view also encompass AD methods.

- *line 355: There is a newer reference available: Moses et al. 2021*

  Noted! Thank you for pointing this out.

- *line 373, 380 (and elsewhere): What you call "backward mode" is usually referred to in the AD community as "reverse mode" (see, e.g., Griewank and Walther 2008).*

  This is correct. Although the two words refer to the same behaviour, we have changed "backward" for "reverse" to stay with the already existing convention. We keep the cases where the use of backwards is semantically more correct, for example "These can be classified depending if they run backwards or forward with respect to the solver."

- *line 389: "the use of AD for optimization..." Not quite right. What you probably mean is " the use of adjoints for optimization". AD is merely a way (albeit a powerful one) to generate code, i.e., the adjoint operator, that efficiently computes the required gradient.*

  This is a good point. We have repharased this to "the use of adjoints for sensitivity analysis integrated with AD tools for optimization and the properties of the landscape generated when using numerical solvers has not" to include the use of adjoints but also emphatized that these also need AD to effectively calculate the adjoints.

- *line 399-401: Sentences: "The gradient calculated by making variation of the parameter $\theta$ capture the variations of $Err(\theta, hyper(\theta))$, which lead to spurious gradients. On the other side, automatic differentiation compute gradients using one single evaluation of $\theta$, meaning that it differentiates only the term $Solver(u0, t0, t1, \theta)$, being robust to the error term." I would love to see this formulated more clearly. As formulated, I find it not easy to understand.*

> We thank the reviewer for rising this point. We have re-write this section to make it more clear and understandable.

- *line 402: obtain − > obtained*

> Noted.

- *Figure A1: Should also define what the points indicated as diamonds are. Also, there exist a classification of numerical discretizations in the atmospheric modeling community, going back to Arakawa and Lamb (1977) Does your grid correspond to any of these?*

> This is a very interesting point. The staggered grid used here corresponds to Scheme E in Arakawa grids, with the difference than here the quantities we evaluate are different (instead of velocities, we evaluate surface gradients; instead of depth, here we evaluate diffusivity). We thank the reviewer for making this interesting connection.

REFERENCES:

Arakawa, A and Lamb, V.R., 1977: Computational design of the basic dynamical processes of the UCLA general circulation model. Methods in Computational Physics: Advances in Research and Applications, 17, 173–265. doi:10.1016/B978-0-12-460817-7.50009-4

Goldberg, D. N., Heimbach, P., Joughin, I. & Smith, B. (2015). Committed retreat of Smith, Pope, and Kohler Glaciers over the next 30 years inferred by transient model calibration. The Cryosphere, 9(6), 2429–2446. https://doi.org/10.5194/tc-9-2429-2015

Griewank, A. and A. Walther, 2008: Evaluating Derivatives. https://epubs.siam.org/doi/book/10.1

Moses, W. S., Churavy, V., Paehler, L., Hückelheim, J., Narayanan, S. H. K., Schanen, M. & Doerfert, J. (2021). Reverse-Mode Automatic Differentiation and Optimization of GPU Kernels via Enzyme. SC21: International Conference for High Performance Computing, Networking, Storage and Analysis, 00, 1–18. https://doi.org/10.1145/3458817.3476165

Schanen, M., Narayanan, S. H. K., Williamson, S., Churavy, V., Moses, W. S. & Paehler, L. (2023). Transparent Checkpointing for Automatic Differentiation of Program Loops Through Expression Transformations. Lecture Notes in Computer Science, 483–497. https://doi.org/10.1007/978-3-031-36024-4$_3$7

---

## Author Response (AR2)

Dear Ludovic Räss,

Thank you for your comments. We have addressed all of them, you can find attached the tracked changes in the full manuscript. The comment regarding the AD properties of Julia had already been removed in the previous version following the comments of Reviewer 2.

Thanks again for your time and help.

The authors